# Lexical phylogenetics of the Tupí-Guaraní family: Language, archaeology, and the problem of chronology

**Fabrício Ferraz Gerardi**[1] *, **Tiago Tresoldi**[2], **Carolina Coelho Aragon**[3], **Stanislav Reichert**[1], **Jonas Gregorio de Souza**[4], **Francisco Silva Noelli**[5]

**1** SfS, Eberhard Karls Universität Tübingen, Tübingen, Germany, **2** Department of Linguistics and Philology, Uppsala Universitet, Uppsala, Sweden, **3** DLPL, Universidade Federal da Paraíba, João Pessoa, Brazil, **4** Department of Humanities, Universitat Pompeu Fabra, Barcelona, Spain, **5** Centro de Arqueologia, Universidade de Lisboa, Lisboa, Portugal

* fabricio.gerardi@uni-tuebingen.de

## Abstract

Tupí-Guaraní is one of the largest branches of the Tupían language family, but despite its relevance there is no consensus about its origins in terms of age, homeland, and expansion. Linguistic classifications vary significantly, with archaeological studies suggesting incompatible date ranges while ethnographic literature confirms the close similarities as a result of continuous inter-family contact. To investigate this issue, we use a linguistic database of cognate data, employing Bayesian phylogenetic methods to infer a dated tree and to build a phylogeographic expansion model. Results suggest that the branch originated around 2500 BP in the area of the upper course of the Tapajós-Xingu basins, with a split between Southern and Northern varieties beginning around 1750 BP. We analyse the difficulties in reconciling archaeological and linguistic data for this group, stressing the importance of developing an interdisciplinary unified model that incorporates evidence from both disciplines.

## 1 Introduction

The problem of establishing the internal relations and chronology of the Tupí-Guaraní language family (henceforth TG) has been a long-standing one. Ideally, there should be a unified model explaining the language expansion and incorporating data from both linguistics and archaeology [1]. The consideration of archaeological data is crucial for establishing the pre-colonial geography of TG populations, which would be very incomplete if based only on historical records, as shown by Fig 1. To achieve it, we began by revising arguments built without considering the archaeological data, especially those developed before the 1960s [2–6], in order to contrast them with other evidence to build our models.

As far as linguistic classifications are concerned, the internal relations of the TG branch of the Tupían family have received much scholarly attention, with different approaches employed to establish them from linguistic data alone. Phonological criteria have been put to use along-side grammatical properties and lexical cognacy, both in "traditional" [7–12] and "quantitative" approaches [13–17]. Although these studies agree to a large extent on the topology of the

**Data Availability Statement:** The supplementary material is available in an anonymous online repository hosted at OpenScienceFramework at the address: https://osf.io/afsyk.

**Funding:** FFG and SR were supported by the by European Research Council (ERC) under the European Union's Horizon 2020 research and innovation programme (Grant agreement No. 834050). TT was supported by Cultural Evolution of Texts project, with funding from the Riksbankens Jubileumsfond (grant agreement ID: MXM19-1087:1).

**Competing interests:** The authors have declared that no competing interests exist.

shallower splits, there are still irreconcilable differences in terms of the deepest ones, and much disagreement about their dating. Previous studies using reduced datasets not designed for phylogenetic analysis [8–10] are still the most commonly referenced ones. The otherwise thorough studies by [11, 12] contained errors in the data that may have influenced the results. [15] is the first Bayesian phylogenetic classification, but neither the underlying data nor the model are public. [18] has several issues, such as low posterior support in branches for well-known cases of relationship (e.g., between Yuki and Siriono, or among Apiaka and Kawahiv languages), analyses including parameters with very low coverage, and the position of some languages (e.g., Kamajurá), besides errors in cognacy judgment. In this study, we make all our data and models available, following the principles of FAIR data [19], and prepare multiple phylogenetic models. Besides providing a phylogeny based on open data, our results are the first to offer a dating of the splits through relaxed molecular clocks. Considering how the question of the root age and the order of splits is a dividing point among specialists, the prospect of building a unified interdisciplinary theory involving linguistic, historical, genetic, archaeological, and ethnographic evidence is considered in the discussion while presenting new groupings.

## 2 Tupí-Guaraní languages and the related archaeology

### 2.1 Languages

TG is the largest branch of the Tupían linguistic family [14, 20, 21], with about 40 living languages (here excluding Piripkura [22]) and at least 9 extinct ones [16]. The number of speakers ranges from less than a hundred (e.g., Amondawa and Juma) to over 6 million (Paraguayan Guaraní) [23]. The geographic distribution, with most TG subgroups found in Southeastern Amazon, points to an origin in this area due to its greater linguistic diversity [24–26]. Such hypothesis contrasts with common interpretations of the archaeological records (pointing to an origin closer to the area between the upper Tapajós and Xingu rivers, further to the west [6]), ethnographic sources, and indigenous cultural repertoire. A clear example of the latter are the foundation myths and legends of the Ka'apor, carrying various hints that they were once located to the west of their present territory [27, 28].

No matter the location of the homeland, the expansion of TG is among the largest in the world, spreading across over 4000 km in both latitude and longitude [33] (see Fig 1), with its driving forces a matter of intense debate [6, 13, 21, 34–42]. Archaeological research suggests that demographic growth was propelled by the rise of agriculture, coupled with a strong sense of territoriality supported by long-range political networks and by an expansionist warlike ideology [6]. An increasing area of forested landscape that could be used for agriculture might have contributed to this expansion [33, 43]. Due to substantial similarities and affinities, material evidence suggests a different scenario and a chronology in line with what one would expect based on linguistic and ethnographic grounds. This is illustrated by the rates of shared cognates, as shown in Fig 2 (also in Fig 7 in Appendix C of S1 File), which are relatively high when compared with those observed in other groups with supposedly comparable dates for their most recent common ancestor, such as Uralic at 43% and Romance at 93% [44]. Archaeological dates considered too ancient have often been discarded, based on the view that the TG dispersal is a recent one. However, over time the accumulation of dates close to ca. 2000 BP in different regions led to a questioning of this premise. Glottochronological estimates of ca. 2500 BP for the initial split of the TG languages [45] have been used to support the archaeological dates. Nonetheless, the discrepancy between such an early chronology and the obvious proximity between the TG languages was never left unnoticed [3, 4].

Any model seeking to explain the evolution of TG needs to account for these facts when proposing language phylogenies [43]. Originally, two such models were put forward. The first

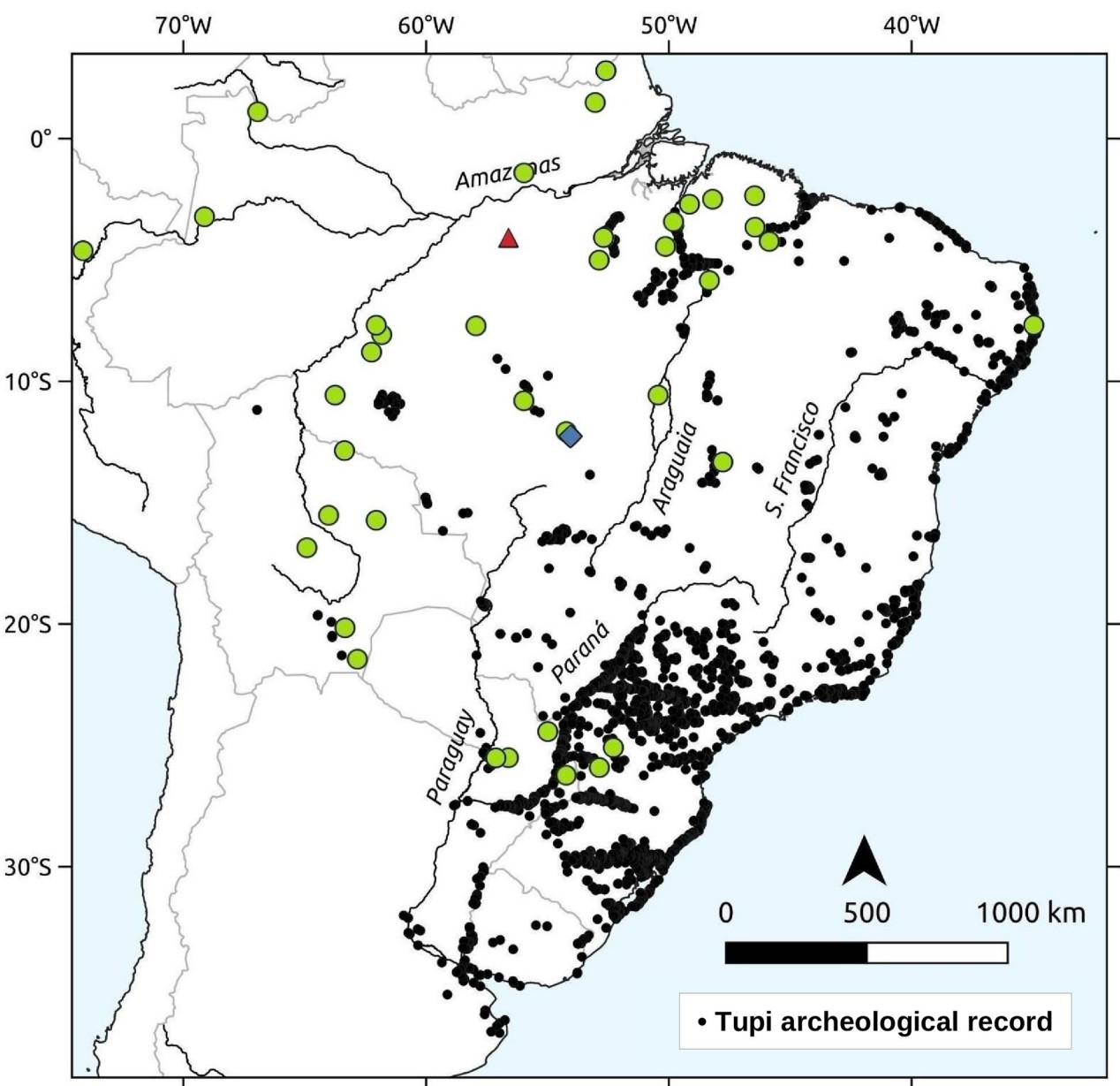

**Fig 1. The Tupí-Guaraní languages used in this study (in green) and the Tupían (non-TG) Awetí (in blue), and Mawé (in red), along with the distribution of the TG archaeological record (black dots).** Prepared by the authors with QGIS 3 [29], based on based on public domain data and raster images from "Natural Earth", including data from [30–32] and an unpublished database by Corrêa and Noelli.

[34] sees the fluvial network as the main enabler of a rapid expansion, an idea further developed by [37, 46], in which the causes of the dispersal are related to climatic factors. The other model finds the key driver in population increase, with the growing need for more cultivation areas (floodplain agriculture) and slower movements of expansion [6, 35, 38, 47, 48].

More recently, a compromise has been found by explicitly testing demographic models against simulated climate change scenarios for the late Holocene [43]. These models show that a combination of demic-diffusion processes and the preference for a particular environmental niche (tropical moist forests) best explains the archaeological chronology and the general

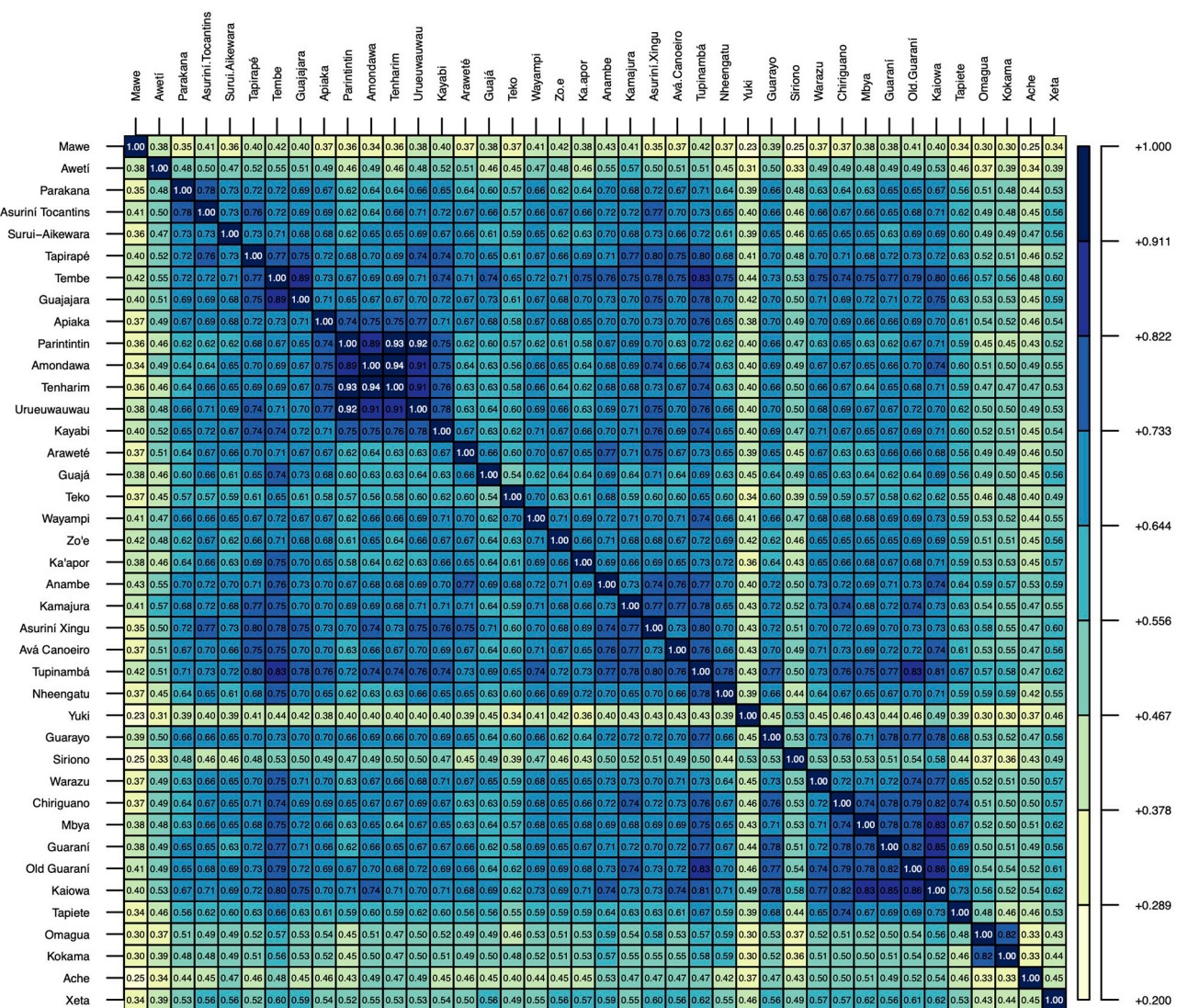

**Fig 2. Cognacy for each language pair used in the main analysis, ranging from 0 (full difference) to 1 (identity).** The cognacy diversity [44] for all languages is 15%. If Mawé and Awetí are excluded, the value is 14%.

reconstruction of historical linguistics: a long stasis in the Amazon, with the emergence and development of the main Tupí branches, followed by a rapid expansion to other parts of South America (corresponding to the TG expansion) [43]. [34] concludes that the most likely center of the dispersion of the TG is the Upper Tapajós. [49] proposes a southwestern Amazonian homeland for Proto-Tupí-Guaraní (PTG), lying near the Arinos and upper Juruena river basins. Using the Linguistics Migration Theory and motivated by the classification in [15, 50] posits the homeland of PTG in the lower Xingu.

The location of the center of expansion is, as expected, dependent of the topology for the family. The Mawé-Awetí-Tupí-Guaraní hypothesis [21, 51–56] states that a single ancestor for these three groups branched off from the rest of the Tupían family [14] (see Fig 3). The split of the branch today composed by Mawé would have been followed by that of the ancestor of Awetí and PTG. Cognates shared by Awetí and TG languages but not present in Mawé support this hypothesis, showing that Mawé had already branched off while the ancestors of Awetí and

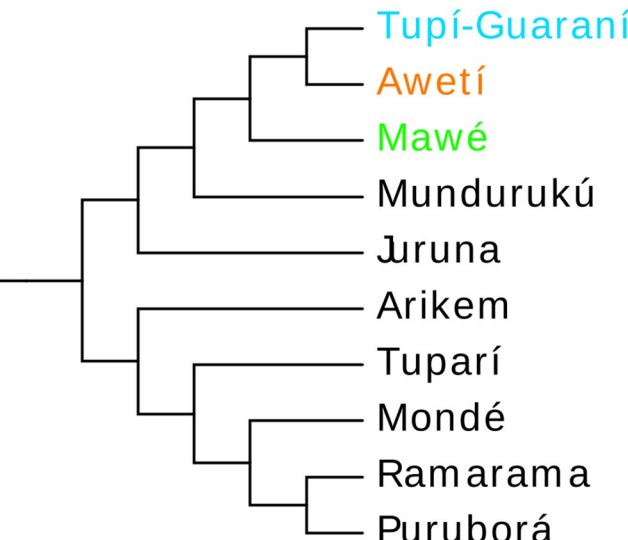

**Fig 3. The Tupían languages with the sub-branches of the Mawé-Awetí-Tupí-Guaraní branch emphasized.** Adapted from [21].

PTG formed single group which heavily borrowed lexical material of Cariban origin [49, 57]. We present some of these cases in Table 1, with cognacy judgment based on [16].

## 2.2 Archaeology

The dispersal of TG languages has a clear material correlate in the spread throughout eastern South America of a package that includes a particular type of ceramics, plant management,

**Table 1. Cognates shared by Mawé and Awetí not present in TG (in yellow) and cognates shared by Awetí and TG (in blue) not present in Mawé.** Tupinambá is taken as a representative of the PTG descendants. The numbers in the last column refer to TG languages whose concepts are cognates with the Tupinambá word provided, illustrating cognates in other branches of TG: Avá-Canoeiro (1), Wayampi (2), Guajajara (3), Parakanã (4), Asuriní Xingu (5), Kamajurá (6).

| | Mawé | Awetí | Tupinambá | |
|---|---|---|---|---|
| Leg | ʔup | ʔup | etɨmã | (1,2,3,5,6) |
| Sing | mepɨ | tepɨ | ɲeʔeŋgar | (2,4) |
| Come back | aipok | ʔajpog | jeβi | (1,2,3,5,6) |
| Hoplias (genus) | (n)ipiuta | piutá | taeʔia | (1,2,3,4) |
| Fly (insect) | win | tin | meu | (2,3,4,5,6) |
| Jaguar | awɨato | tawat | jawa | (1,2,3,4,5,6) |
| Anteater | arihĩ | tamajua | tamandwa | (1,2,3,4,6) |
| Grandfather | aseʔi | amũj | amĩja | (2,3,4,5,6) |
| Wound | pihi | peʒep | peɾeb | – |
| Tapir | wewato | tapiʔit | tapiʔiɾ | (1,2,3,4,5,6) |
| Sky | atipɨ | ɨwak | ɨβak | (1,2,3,4,5,6) |
| Genipa | wããhop | tẽtɨpap | janɨpab | (2,3,4,5,6) |
| Bat | hakiʔi | tatiʔa | anɨra | (1,2,3,4,5) |
| Sieve (tool) | panane | kurupem | urupem | (2,4,5,6) |
| Burn (something) | wuk | apɨ | apɨ | (1,2,3,4,5,6) |
| Bow | moreawat | ʒapat | ɨβɨɾapaɾ | (1,2,3,4,5,6) |
| Star | wajkiru | tatɨaʔɨt | jasɨtata | (1,2,3,4,5,6) |

and cultivation of a variety of crops [6, 38], as shown in Fig 1. This is often cited as one of the few cases where an obvious correlation exists between an archaeological culture and a language family, to the point where the name "Tupiguarani" (no hyphen) was applied to the archaeological tradition (for a criticism of this concept see [47, 58]). Admittedly, correlating a material culture style with the speakers of a single language or language family is in most cases a problematic, if not naïve, approach. Similarly, material culture changes may precede or postdate related changes in society and language [59–61]. Nevertheless, there is overwhelming evidence to support the association between the ceramics conventionally called "Tupiguarani" and the spread of the Tupí-Guaraní language family. Of particular interest is the notable homogeneity of the material culture throughout the Tupí-Guaraní territory [31]. This conservatism is seen even in areas historically occupied by linguistically distinct groups such as the Tupiniquim and Tupinambá [62]. The high standardization in ceramic styles across time and space—accompanied by the maintenance of a specific vocabulary to describe vessel shapes [63]—is a testimony to the conservatism found in other spheres of the Tupí-Guaraní cultures [64]. Ultimately, the ceramics recognized as "Tupiguarani" by archaeologists can be traced back to the Tupían homeland in southwestern Amazon, where its stylistic components, such as polychrome painting, can be found among other ceramic traditions [65].

In what follows, we summarize the earliest radiocarbon dates available for Tupiguarani sites. The dates are divided according to five regions: Eastern Amazon, Bolivia, Atlantic Coast, Northern Brazil, and the Paraná Basin. All dates are calibrated with the southern hemisphere curve [66] and reported in the 2-sigma interval.

**Eastern Amazon** An early presence in the Xingu-Tocantins interfluve is supported by the available radiocarbon dates. A date of 2430 ± 20 BP (cal BP 2680–2340) from a site in the Tocantins-Araguaia confluence is still seen with caution, as it is considerably older than all other dates from the same region [67, 68]. The accepted Tupiguarani chronology for the eastern Amazon starts at 1670 ± 80 BP (cal BP 1700–1350) between the Tapajós and Tocantins rivers [68].

**Bolivia** The earliest potential TG site in pre-Andean Bolivia has a date of 1680 ± 90 BP (cal BP 1730–1320, UA-10240), which, if confirmed, would imply an arrival of the Guaraní-speaking Guarayo and Chiriguano in the region earlier than commonly thought [69].

**Atlantic coast** In the region historically occupied by the Tupinambá, a controversial early chronology has been proposed by Scheel-Ybert et al. [70], based on dates reaching 2920 ± 70 BP (cal BP 3220–2790, Gif-11045) from sites in the state of Rio de Janeiro. These predate the TG expansion out of the Amazon by any estimate. Excluding those outliers, the earliest date for the Atlantic forest is of 1740 ± 90 BP (cal BP 1825–1380, Beta-84333) [70], which is in line with the chronology of other parts of the TG territory. Most dates are considerably more recent, later than 1055 ± 80 BP (cal BP 1060–740, SI-828) [71].

**Northeastern Brazil** Few dates are available for northeastern Brazil. In the semi-arid hinterland, a date of 1690 ± 110 BP (cal BP 1810–1315, GIF-3225) is sometimes attributed to a TG occupation, but the cultural affiliation of the dated site is not a consensus [38, 72]. Discounting dates with excessively large standard deviations [73], the occupation of the coast possibly extends back to 1880 ± 60 BP (cal BP 1920–1590, Beta-118818) [31], with most dates being considerably later.

**Paraná Basin** The southernmost region of Tupí-Guaraní occupation, where Guaraní and related languages were dominant, has the most complete and reliable chronology [32]. The earliest date, 2010 ± 75 BP (cal BP 2090–1740, SI-5028), comes from the middle Paraná river [74]. Between that date and the second millennium, multiple sites are attested in the São Paulo highlands, southernmost Brazil, and the Paraná-Uruguay interfluve in Argentina [74].

## 3 Materials and methods

### 3.1 Data

We followed the current best practices for linguistic phylogenetics ("phylolinguistics"), where cognate gain and loss in basic vocabulary are the evolutionary characters used to infer a dated tree [75–77]. The complete dataset used in this study is derived from [16] and is publicly available, along with the phylogenetic models, at https://osf.io/afsyk. For better integration with other linguistic resources, we standardized the data following the formats and catalogues of [78]. Cognate set assignment, following the principle of root-meaning traits [79], was first performed with the automatic methods implemented in LingPy [44, 77, 80, 81] and later manually reviewed by experts in its entirety. Table 2 shows a sample of cognacy assignment from [16].

The data in our study comprises lexica from 40 "doculects" [82] (i.e., language varieties). Mawé and Awetí were included in the analyses, with the split of the Mawé ancestor serving as the root and reflecting the aforementioned and well established Mawé-Awetí-TG hypothesis. We also included Omagua and Kokama due to a high portion of their lexicon being of TG origin, despite their non-TG origin [83, 84], a hypothesis rejected by [85]. Some TG languages available in our source were excluded from the analyses due to an excessively low coverage.

The list of concepts is provided in Appendix A of S1 File, along with the corresponding Concepticon cognate set ids and glosses [86] when available. The choice of concepts relied on the following criteria: concepts from the Swadesh [87] and Leipzig-Jakarta [88] lists, the Swadesh list extended by [89], and culturally relevant TG concepts taken from [90] and expanded by the authors. The concept coverage for each language is given in Table 3. We used 415 concepts from an upcoming version of [16]. We assessed the degree of tree-likeliness by computing the concepts' TIGER scores [91, 92] with the implementation by [93], obtaining a mean score of 0.14 (±0.14) (individual scores are reported in Appendix K of S1 File). This value suggests a comparatively high level of non-vertical transmission, being lower than the lowest score reported in [93] of 0.20 for Dravidian, and supports the qualitative assessment that "there is an overall absence of well-delimited lexical clusters inside [TG]" [13].

### 3.2 Phylogenetic reconstruction and dating

Data was prepared with the state-of-the-art software tools for computer-assisted pipelines in computational historical linguistics [80, 94] and exported in the extended NEXUS format [95]. The files produced by this pipeline were processed and normalized with Python scripts developed for this research.

Since the evolutionary history of the TG languages is not completely tree-like, as per [13] and measures in Section 3.1, we first generated a distance matrix to build a NeighborNet network using SplitsTree version 4.17.1 [96] to visualize the conflicting signal and calculate the $Q$-residuals and the $\delta$-scores.

Different phylogenetic models were then explored in terms of subsets of concepts, languages, molecular clocks, calibration dates, substitution models, rate variation, and

**Table 2. Cognacy sample from our database.**

| Language | Concept | Phonetic form | Cognate set |
|----------|---------|---------------|-------------|
| Tupinambá | BAT | anɨra | 171 |
| Wayampi | BAT | anɨla | 171 |
| Guaraní | BAT | mopi | 172 |
| Kaiowá | BAT | ᵐbopiri | 172 |
| Mawé | BAT | hakiʔi | 4513 |

**Table 3. Concept coverage for the languages used in this study from [16].**

| Language | Glottocode | ISO 639–3 Code | Coverage |
|---|---|---|---|
| Ache | ache1246 | guq | 80% |
| Amondawa | amun1246 | adw | 74% |
| Anambe | anam1249 | aan | 49% |
| Apiaka | apia1248 | api | 65% |
| Arawete | araw1273 | awt | 71% |
| Asuriní Tocantins | toca1235 | asu | 84% |
| Asuriní Xingu | xing1248 | asn | 63% |
| Ava-Canoeiro | avac1239 | avv | 79% |
| Aweti | awet1244 | awe | 93% |
| Chiriguano | east2555 | gui | 90% |
| Guaja | guaj1256 | gvj | 80% |
| Guajajara | guaj1255 | gub | 97% |
| Guaraní | para1311 | gnn | 99% |
| Guarayo | guar1292 | gyr | 89% |
| Ka'apor | urub1250 | urb | 95% |
| Kaiowa | kaiw1246 | kgk | 51% |
| Kamajura | kama1373 | kay | 68% |
| Kayabi | kaya1329 | kyz | 63% |
| Kokama | coca1259 | cod | 82% |
| Mawe | sate1243 | mav | 88% |
| Mbya | mbya1239 | gun | 84% |
| Nheengatu | nhen1239 | yrl | 91% |
| Old Guaraní | oldp1258 | grn | 83% |
| Omagua | omag1248 | omg | 80% |
| Parakanã | para1312 | pak | 83% |
| Parintintin | tenh1241 | pah | 96% |
| Siriono | siri1273 | srq | 94% |
| Surui-Aikewara | suru1262 | mdz | 83% |
| Tapiete | tapi1253 | tpj | 77% |
| Tapirape | tapi1254 | taf | 68% |
| Teko | emer1243 | eme | 96% |
| Tembe | temb1276 | tqb | 93% |
| Tenharim | nucl1663 | pah | 76% |
| Tupinamba | tupi1273 | tpw | 99% |
| Urueuwauwau | urue1240 | urz | 60% |
| Warazu | paus1244 | psm | 85% |
| Wayampi | waya1270 | oym | 99% |
| Xeta | xeta1241 | xet | 61% |
| Yuki | yuqu1240 | yuq | 64% |
| Zo'e | zoee1240 | pto | 82% |

monophyletic constraints. We decided in favor of the simplest and most common practices whenever possible and sensible, following the principle that we should begin with more approachable studies before venturing into more complex scenarios. The initial exploration, partly published in [97], was relevant for the authors to discuss the concepts that were deemed less reliable, and the problems that could arise from the analysis. These studies also served to evaluate the feasibility and robustness of our approach.

We structured the research into two rounds, the first one designed to obtain summary trees given different scenarios of analysis and the second one using these results to perform a phylogeographic study. The first round was composed of two studies that differ in the subset of concepts used: a "full" study, with all concepts described above filtered to ensure they were missing at most in 20% of the languages, and a "swadesh" study using the list of [87] (see Appendix C in S1 File) as close as possible, filtered to ensure they were missing in at most 30% of the cases. Such thresholds were necessary due to the high level of sparsity of the data. In both cases concepts were grouped in two equal-sized partitions based on the overall number of cognates in each. Besides simpler strict-clock models, which are comparable to glottochronological approaches, all analyses also used uncorrelated relaxed-clock models sampled from a lognormal distribution [98, 99]. In the latter, each branch of a tree has its own clock rate, with parameters that are independent from those of the mother and sister branches, allowing abrupt changes in evolutionary rates. These are considered compatible with both the evolution of TG, given its relatively recent and rapid expansion, and South American languages in general, particularly due to the impact of European colonization in terms of population size, displacement, and replacement [100, 101].

We performed phylogenetic reconstruction using BEAST2 version 2.6.6 [102], fitting different binary covarion models [103], where the transition between "presence" and "absence" of a cognate in a language is assumed to be symmetric and equally probable, along with a latent variable modeling whether each cognate switches between presence and absence at a "fast" or "slow" rate. Ascertainment correction was performed according to practices described in [76]. Considering how our data only offers two historical languages that could be used for temporal calibration (Tupinambá and Old Guaraní), both of which are to some extent composed from multiple sources diverging in provenance and date (each spanning over more than a hundred years), we decided to guide the inference only by setting a uniform distribution for the root, in agreement with all sensible archaeological and linguistic hypotheses, and by establishing monophyletic groups accepted by virtually all experts, also adjusting tip dates for languages collected more than 50 years ago (detailed in Appendix D of S1 File). We used a Birth-Death model [104], performing $25^7$ MCMC iterations, sampling trees from the posterior distribution to obtain a maximum clade credibility tree (MCC) based on common ancestor heights after a 50% burn-in, using TreeAnnotator version 2.6.4 [105]. We plotted trees with [106] and FigTree version 1.4.4; the trees, including for the supplementary models, are presented in Figs 8–11, all in Appendix E of S1 File.

The results in Section 4 are those of the "full" study using a relaxed clock. The decision in favor of this model as our main result is based on the set of concepts it involves, which, despite a higher reticulation signal, includes family-specific concepts that were deemed relevant for studying the vertical transmission. It is necessary to note that the logmarginal likelihood (see Appendix L in S1 File), computed with nested samples [107], not only favored the "swadesh" dataset, as expected in face of its lower data complexity, but it also yielded a better score for the strict molecular clock in the case of the "full" dataset. Our decision in favor of the relaxed clock model was due to an expert analysis of the resulting topology and dates, as it was far more compatible with the literature, and by the fact that most unexpected results, such as the position of the Anambé-Araweté clade or the branch length of Tupinambá, can be explained by differences in concept coverage. The complete studies are presented in the supplementary material and should guide future research and refinements to cognate judgments.

The phylogeographic study used the topology of the MCC tree of the "full" study as a set of monophyletic constraints wherever we had obtained a posterior support of at least 0.70, along with the 95% height range for each such split, focusing on having the model search for dates

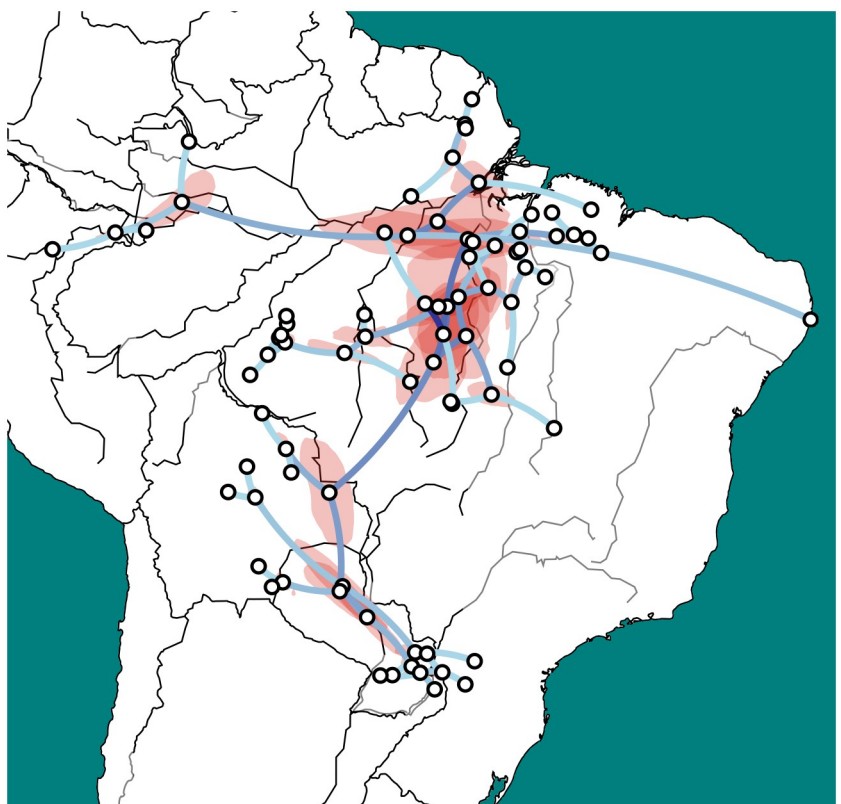

**Fig 4. Output of the phylogeographical model.** Brightness of edge colors (blue shades) indicates the mean common ancestor height, with darker colors indicating older inferred movements. Geographic areas in red indicate the 80% confidence for location of intermediate nodes. An interactive visualization is available online at https://tupiguarani. netlify.app/ and in the supplementary material. Prepared by the authors with SpreaD3 version 0.9.6 [109], based on public domain data and raster images from "Natural Earth" for political boundaries and hydrography.

and geographic locations only. It used the GEO_SPHERE model version 1.3.1 [108], building the visualization in Fig 4 with SpreaD3 version 0.9.6 [109] on top a politico-hydrological GEOJSON map of South America prepared by us.

All models were also investigated using Densitree version 2.2.7 [110, 111] to visually identify conflicts and signals compatible with non-tree evolution (as evidenced by the one provided in Appendix F of S1 File).

## 4 Results

### 4.1 NeighborNet network

The neighbor network (NN) for the group is given in Fig 5. The $Q$-residual value (0.005957) and $\delta$-score (0.3861) for the whole family are comparable to the values listed for other languages in [8, 112]. The $\delta$-score is a measure of the tree-likeness of phylogenetic distances before the estimation of the tree, that is, it identifies how much a taxon is involved in conflicting signals (different possible evolutionary trajectories) [113]. The $\delta$-scores are estimated in terms of four taxa (quartet). The $Q$-residual [113–115] is a type of measure over all values in the quartets [114]. The quartets are the boxes seen in a NeighborNet like Fig 5.

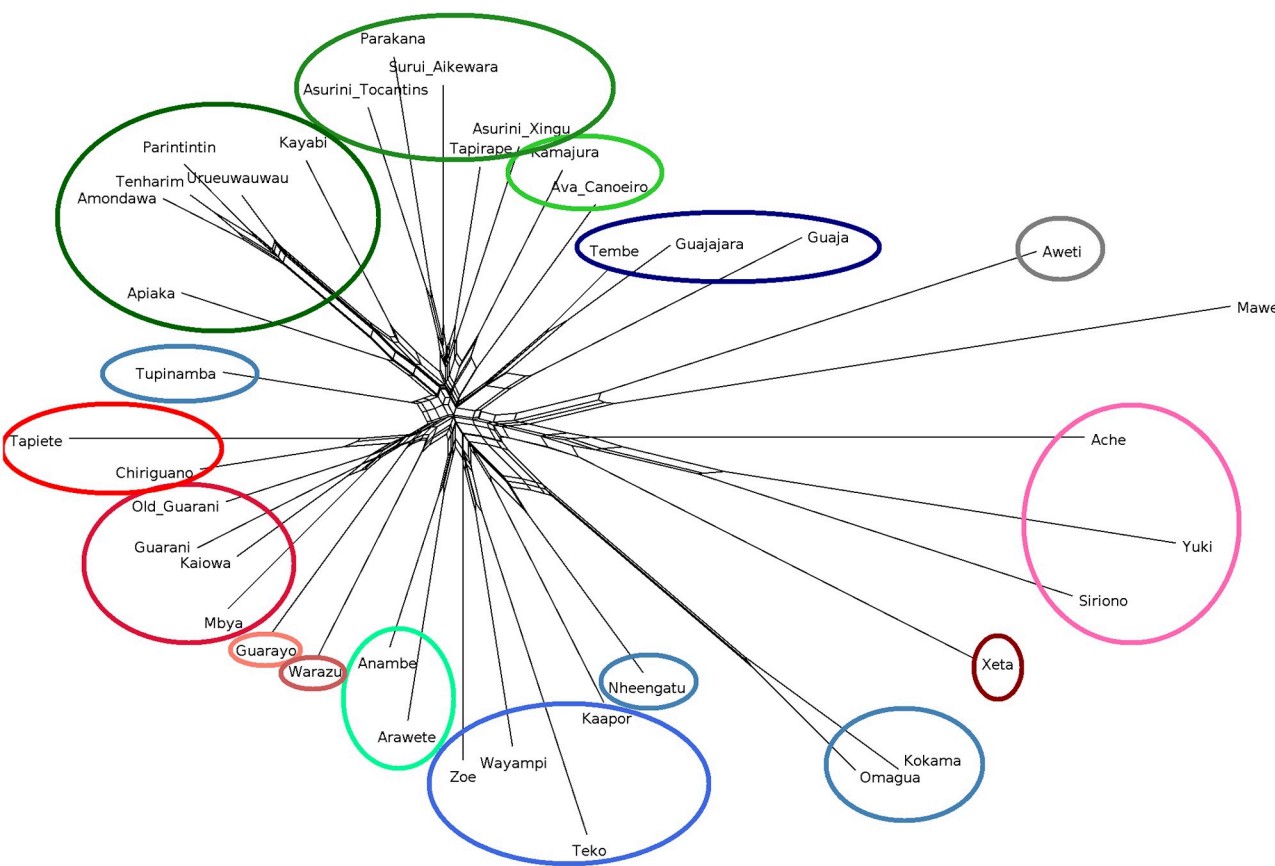

**Fig 5. NeighborNet illustrating the reticular relationships from the data used in the study, built using rates of shared cognacy.** The colors correspond to the groups in Fig 6.

## 4.2 Tree topology and dating from phylogenetic reconstruction

The MCC tree resulting from the study is shown in Fig 6. According to it, Mawé separates from its ancestor about 3300 (95% HPD: 2500–4620) years ago, while Awetí separates about ca. 2600 (95% HPD: 1404–4037) years ago. It is only at around 1700 BP (95% HPD: 847–2740), after a stable period of about 800 years, that the Tupí-Guaraní group begins to spread. Two major splits separate the ancestors of groups I, II, III, as defined and described in Section 5; date estimations for the most important splits are reported in Table 4.

## 5 Discussion

The NN is compatible with claims of a recent arrival of TG to the coast and particularly with a relatively high overall admixture (such as in the reticulation between Ka'apor and Nheengatu, Nheengatu and Kokama-Omagua, the Kawahiv languages, and the Suruí Aikewara-Parakanã-Asuriní Tocantins clade). Mawé and Awetí, whose structure tends to be confirmed by shared lexical innovations between Awetí and TG, share lexical material that is not found elsewhere in TG and which would be more compatible with a common non-TG source. Siriono and Yuki share a signal compatible with hybridization between the ancestors of Ache and Xeta, an observation that can be extended to Guajajara (showing a signal compatible with a hybridization between the latter and Guaja), and to the Urueuwauwau-Parintintin-Tenharim-

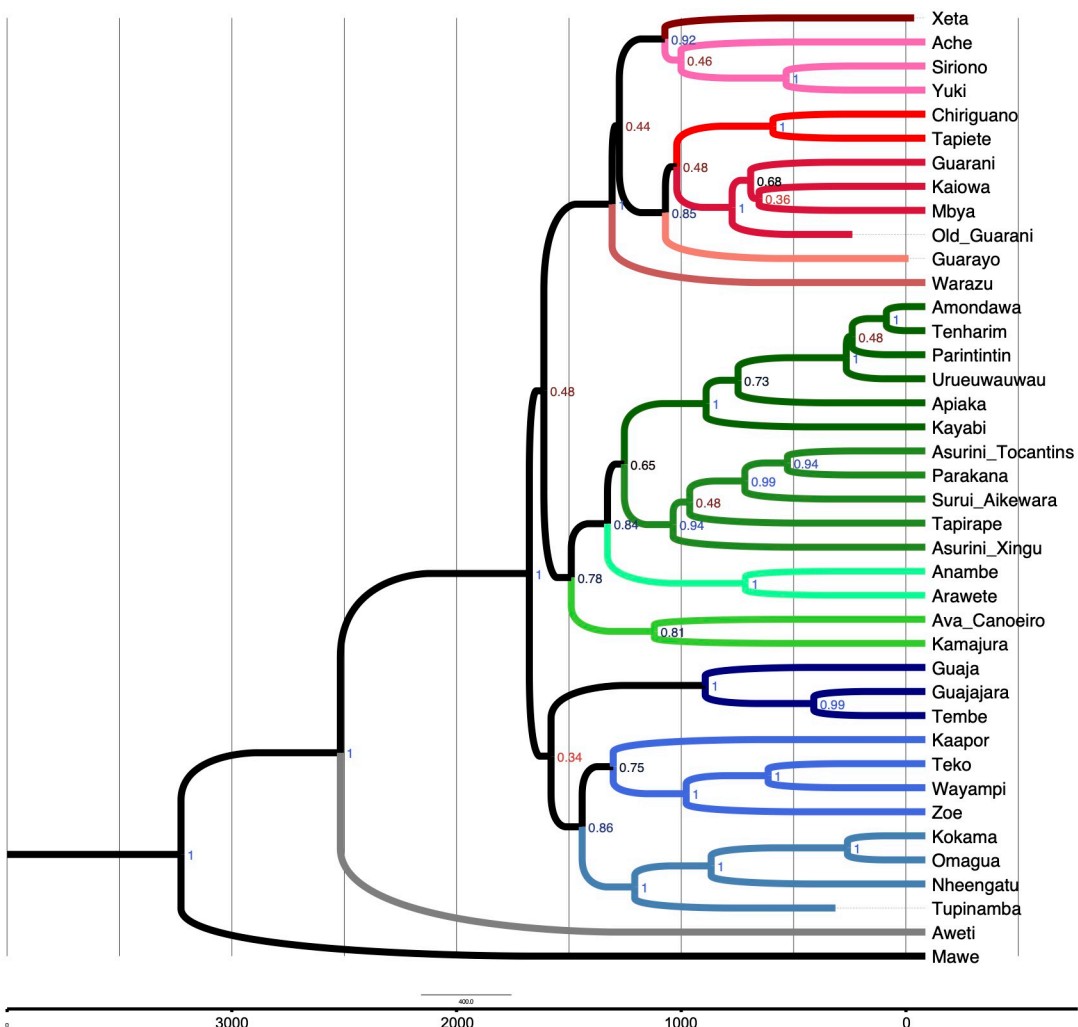

**Fig 6. The maximum clade credibility tree from the "full" model.**

Amondawa clade (showing a signal compatible with a Kayabi-Apiaka hybridization). The strong distinctive signal of differentiation of Kokama and Omagua is confirmed (potentially supporting [85]), with Nheengatu being the closest but, nonetheless, a distant relative. The NN also highlights issues with our data, such as the position and relative long branch of Tupi-nambá in relation to its known descendant Nheengatu, in part also reflecting the numerous lexical contributions from this branch into many different groups.

The MCC tree shows dates that are rather close to those suggested by archaeological studies (see Section 2.2) and in particular [116], who places Proto-Mawé-Awetí-TG in the region of the Tupinambarana Island around 2500 BP and Proto-Awetí at the high Xingu Basin in the 2100 BP. After a stabler period, compatible with theories of punctuated equilibrium in language evolution [117], at approximately 1750 BP a major split divides the TG branch in two major clades, with a further division of one of these groups. The low posterior values of such splits (0.34, 0.48, and 0.44, respectively) and their temporal proximity are compatible with the scenario of a hard polytomy suggested by archeological hypotheses of a rapid radiation. One split involves the ancestor of all the TG languages spoken in southern Brazil, Paraguay, Bolivia,

**Table 4. Node height and 95% HPD for the most important splits in the tree.**

| Split | Node height (YPB) | 95% HPD (YBP) |
|---|---|---|
| Mawé / Awetí-TG | 3312 | 2500–4620 |
| Awetí / TG | 2603 | 1404–4037 |
| TG disintegration | 1762 | 847–2740 |
| Group I | 1665 | 842–2476 |
| Group II | 1575 | 721–2329 |
| Group III | 1394 | 811–2561 |

and Argentina (group III), and the TG languages that remained closest to the TG putative homeland in the Xingu-Tapajós interfluve (group II). The other group consists of languages that moved away from the homeland (group I).

By combining quantitative results, previous linguistic classifications, and ethnographic literature, we can propose three major language groups ("clades") that can guide future discussions and research. These are colored in our tree in blue (group I), green (group II), and red (group III). The different shades of each of these colors indicate subgroups, and are:

**Group I**, which is divided in subgroups Ia, Ib, and Ic according to the order of branching. The whole group is characterized by dispersals that brought its members further away from the Proto-Tupí-Guaraní homeland.

Subgroup Ia comprises Tembé and Guajajara (Tenetehara), and Guajá. It should be no surprise that Guajá (Rodrigues' group VIII) clusters with the Tenetehara languages (Rodrigues' group IV), since their location is at an intersection zone, a reason why Guajá and the Tenetehara languages have a high rate of shared cognacy that includes important disjunctive innovations. In fact, Guajajara (73%) and Tembé (74%) show the highest rates of shared cognacy with Guajá. The Guajá have been reported for at least 150 years close to the Pindaré, Turiaçu, and Gurupi rivers, in contact with the Ka'apor, Tembé, and Guajajara. The upper Pindaré river has been home to the Tenetehara since they are first mentioned in 1615 [118]. Its proximity with the Tenetehara languages may not necessarily be due to shared inheritance, but nothing is known about the Guajá previous to the contact [119, 120].

Subgroup Ib is composed of Zo'e, Wayampi, Tekó, and Ka'apor, paralleling Rodrigues' group VIII. Ka'apor is not only phonologically close to Wayampi and Tekó, as shown in [10], but its speakers are also culturally related to Wayampi, as shown by [27, 121].

Subgroup Ic comprises Kokama, Omagua, Nheengatu, and Tupinambá. Tupinambá and Nheengatu are placed in Rodrigues' group III, while Kokama and Omagua are not listed among TG languages, not even in [21]. One relevant issue in this subgroup concerns their status, being considered either the descendants of a non-TG language which acquired TG lexicon [83, 84], or a pre-Columbian language, product of the contact with a TG language by [85]. The question cannot be solved by cognate sets alone, and what concerns us here is the fact that Kokama and Omagua belong to the same clade as Tupinambá and its descendant Nheengatu.

Regarding the proximity of Ka'apor with subgroup Ic, it can be explained by its many lexical borrowings from Língua Geral [52, 122], as captured both in the density tree (Fig 12) in Appendix F of S1 File, where the conflicting signals approximate it to subgroup Ic, and in the MCC 0.75 posterior support. Its proximity to Zo'e owes to the fact the latter does not share some innovations present in Wayampi and Tekó. The Wayampi are known to have lived in the Lower Xingu, where the Ka'apor were once located [28, 121]. [121] even mentions that, according to Ka'apor informants, they could understand Wayampi better than other any TG language they had heard.

**Group II** is formed by the languages that remained closest to the postulated PTG home-land. The Kawahiv group is no exception, since it is known to have migrated towards Rondô-nia only in the nineteenth century [123, 124]. The group is internally organized as follows: Avá-Canoeiro and Kamajurá (IIa); Anambé and Araweté (IIb); Asuriní Xingu, Tapirapé, Suruí-Aikewara, Parakanã, and Asuriní Tocantins (IIc); Kayabi, Apiaka, Parintintin, Urueu-wauwau, Tenharim and Amondawa (Kawahiv clade) (IId).

Avá-Canoeiro and Kamajurá (IIa) have a relatively medium coverage in our database (78% and 68% respectively), but one does not need to take the clade with these two languages as improbable, as the analyses under the "swadesh" model also groups them together. They also show up relatively close in the classification in [18], where their coverage was significantly smaller than in the current analyses. Little is known about the Kamajurá (Rodrigues' group V), except that they might have entered the Xingu area in the second half of the eighteenth century [125, 126]. The Canoeiros (Rodrigues' group IV) were reported at the head of the Tocantins river in the 1700s, [127] with subsequent movements fairly well documented: to the Araguaia region in 1830, later towards the state of Pará, and finally towards the Javaé, their current seat, before the 1900s [128]. If these groups ever were in contact, it must have been long ago, some-where between the lower Xingu and the lower Tocantins, where they were part of a larger group associated with the other languages of our group II: certainly before the eighteenth cen-tury, even though it is currently impossible to determine any date with certainty.

Subgroup IIb comprises Araweté and Anambé (Cairari), both grouped together in [10, 11]. Note that the latter is not the homonym language from the wordlist by Ehrenreich [129], which belongs to Group VIII in [10]. As suggested by [64], whatever is said about the Araweté before the contact is nothing but a conjecture, and the situation is not different for Anambé [130]. Both languages are also grouped together in [21] (group V). The proximity of Anambé with Araweté has also been stressed by [130] and by [131], who assert that Araweté shares more linguistic similarities with Anambé and Asuriní Xingu than with any other language.

Regarding subgroup IIc, Asuriní Xingu and Tapirapé are part of a binary branching in [11]. The Tapirapé, which were part of the group that remained in the North, have indeed once been at the interfluve of the Tocantins-Xingu [132]. Their journey southwards is probably related to constant conflicts with the Kayapó and Karajá [132]. The Suruí-Aikewara have moved little since the group split: more likely than not, they are the people described by [133] in 1898 as living along the Itacaiúnas and Araguaia rivers, near the Tocantins banks. In 1904 they were located close to the head of the Sororó river [134]. This is consistent with a putative eastward movement. The Asuriní Xingu are reported for the first time at the Bacajá river in 1894 [135].

Subgroup IId has members that not only speak similar language varieties, but which are also culturally homogeneous [136–138]. There is little doubt that these languages belong to a super clade with IIb and IIc; for example, Amondawa shows 74% of cognate agreement with Asuriní Xingu. Their migration towards the Upper Madeira river is known to have happened relatively later, during the colonial period [10, 123, 139–143]. They were first located at the Upper Tapajós and subsequently at the Middle Machado [144]. Although Asuriní Tocantins and Parakanã are considered a dialect group by [138], the inclusion of Suruí-Aikewara in the group is not controversial.

**Group III**'s internal organization is: Warazu (IIIa); Guarayo (IIIb); Old-Guaraní, Mbyá, Kaiowá, and Guaraní (IIIc); Tapiete and Chiriguano (IIId); Xetá (IIIe); and Yuki, Sirionó, and Aché (IIIf).

Warazu as a single-language subgroup reflects Dietrich's assertion that it is a language inde-pendent of all others [124], an assumption supported by the full posterior value for this split. The split resulting in a single clade with Guarayo has high support (0.85). In fact, Guarayo

seems to share some characteristics with Old Guaraní (or with its ancestor) [124] not observed in any other language. Its position in the tree also reflects the idea of a single origin postulated by [145].

[124] likewise identifies a Chiriguano-Tapiete subgroup (Rodrigues' group I), describing Tapiete as a dialect of Chiriguano, reflected by the full posterior support in our tree. [32] discuss the similarities between these languages as well, showing that phonological properties corroborate the separation of Chiriguano-Tapiete from other languages.

According to Rodrigues [146], Sirionó and Yuki are subgroups of the dispersion of Guarayo and Warazu. This is a possible scenario according to our tree. Nonetheless, both former languages would be expected to appear in a clade with other "Guaraní" languages, if the source from which they adopted TG elements (lexical and grammatical) was either Old Guaraní or a language variety related to it [124]. The Guarayo and Warazu are similar not only in language, but also in culture, both differing from the Sirionó [147].

Aché [148] and Xetá [149] are languages that recently went through a process of Guaranicization [9, 124]. Due to the low coverage for these two languages, among the lowest in our dataset, we refrain from further conjectures. Their position as outliers within the family is however not controversial. [150] follows the hypothesis that the Warazu might have come from the upper Tapajós river to the Guaporé, affirming that the name Guarayu (an ethnonym related to the Warazu for many years, which apparently still leads to confusion in [151]) is found in two discontinuous areas: from the Guaraní area to Bolivia and in the Tocantins region. When discussing the migration of the Guarayo, [146] locates them further to the Paraguay river, towards the northeast and later towards the Amazonian basis. One part of the group would have remained along the Paraguay river, proceeding southwards, being the ones described by European sources in the 1700s and 1800s.

Among its main findings, our topology, besides supporting recent genetic studies that favor a north-to-south colonization of the coast [152] contrary to [49], showed that the Tupinambá are linked to the "Amazonian" group. This Amazonian group would have take a different part from the ancestor of Guaraní, once more contradicting [49]. In terms of differences with the previous phylogenetic classification by [15], we decided to withhold from deeper comparisons as neither their model nor their data are available. In terms of topological disagreements, we favor our tree due to a number of groupings that are less problematic and questionable. For example, [15] cluster Ka'apor with Guajá and Avá-Canoeiro, despite it sharing only 64% of its cognates with both these languages, against rates of 73% with Tupinambá and 74% with Tembé. On the other side, Avá-Canoeiro and Kamajurá share 77% of their cognates and the two Kawahiv languages in their sample are closer to Tembé and Wayampi than to Asuriní Tocantins. The amount of cognates between the Kawahiv group and other languages of group II is significantly higher than with Tembé or Wayampi. As attested by [153], the Kawahiv were once located between the Tapajós and Xingu rivers, thus closer to the Asuriní Tocantins, Parakanã, and Suruí than to the groups there suggested. Historically, the Kawahiv languages have been long separated from Wayampi, Tekó, since these have been at their current locations for centuries [154–157], with Tembé likewise already at their current location at least the beginning of the seventeenth century [118, 158]. Another perceived shortcoming concerns the proximity of Tupinambá with the southern languages, in opposition to the aforementioned genetic studies. The branching order is also difficult to accept in light of our historical knowledge on some the languages, but even this judgment is limited in the absence of data which is described but not provided.

The analyses presented here do not deviate significantly from [18], which used different models. The main differences can be observed on the lower branches while there is considerable agreement as far as the higher branches and sub-groups are concerned.

## 6 Concluding remarks

Most cases of lower posterior support in our tree can be explained, at least in part, by missing data. The low confidence in some of the splits within individual groups, such as among the Guaraní languages, might be due to both technical aspects, like an unequal level of sampling, and the actual history of the languages, involving dialect chains, admixture both at the linguistic and genetic levels, etc. These issues are heightened by the lack of calibration data of temporal and geographic matter that can be applied directly, as well as by our decision to begin this research path by using models which are simpler to understand and less susceptible to prior hypotheses specified by us. The topology and the posterior support are expected to improve as we extend the data, employ more complex models (which tend to involve different types of calibrations), and, potentially, the direct or indirect usage of additional linguistic evidence to allow the a priori definition of monophyletic groups, aiming for more precise parameters of local evolution. The historical-anthropological survey work presented in the previous sections, in particular, may prove to be extremely valuable in future research, provided that it is used with the due caution (see [159]).

It is essential to emphasize that the classification presented here is exclusively based on lexical changes, although for most of the clades there is a significant agreement with Rodrigues' taxonomy based on phonology [10], and even with the cultural classification in [42]. A caveat is necessary here: linguistic classifications based exclusively on phonological changes, such as the one by [10], are generally considered to be more susceptible to common independent innovations, that is, cases in which the same character (a sound change) independently arises more than once in different branches, leading to "homoplasy". This is one of the main reasons for the suggestion that most phylolinguistic studies should involve exclusively or majorly characters based on lexical innovations, which can be assumed to be independent and arise only once. Likewise, when considering differences with archaeological datings, it is worth noting that such phylolinguistic models consider, and by extension date, splits as the moment when the first disjunctive lexical innovation in the basic vocabulary takes place. Such event does not necessarily imply a degradation of mutual intelligibility, nor can it be automatically associated with either population displacement or changes in archeological packages.

In terms of routes of expansion, we believe that the ancestors of Tupinambá took different directions, traveling eastwards, while the rest of the group traveled westwards first and then northwards. Paralleling Rodrigues' Group VIII (which includes Guajá), the group containing Ka'apor, Tekó, Wayampi, and Zo'e is clearly supported by phonological and lexical innovations alike, despite the presence of the Tekó in the French Guiana already in the 1500s [160]. Since Zo'e is phonologically closer to Tekó than to Wayampi [161], it is possible that the ancestors of the Zo'e, as those of the Tekó, had already separated from the ancestors of Wayampi, whose migration northwards from the lower Xingu river only began in the early 1600s [154, 155, 162]. The late split of Wayampi and Tekó in our tree is probably caused by innovations common to both groups and borrowings of Cariban origin not present in Zo'e, exemplified in Table 5. It is unknown whether the Tupían group referred to as "Apama" [144, 163] and described in 1691 between the Curua and Maicuru are ancestors of the Zo'e, who in 1600 were still located in Lower Xingu. If the identification of this group with the Zo'e is correct, we could infer their movements based on additional, non-linguistic evidence.

The migration of multiple groups towards Rondônia during the colonial period is not only acknowledged by multiple sources [123, 139, 140, 143, 164], but can also be demonstrated linguistically on the basis of the abovementioned Carib loans in PTG [124], not found in Mawé or Awetí. These loans are also found in the Kawahiv languages. Of all conjectures regarding how the Kamajurá reached their current location [126, 132, 165], an attractive one is told by

**Table 5. Lexical innovations in our Group Ib (Wayampi, Tekó, and Zo'e).** Some, not shared by Zo'e, took place when Wayampi and Tekó were already in the French Guiana, as the source of the borrowings indicates. The word for 'timbo liana' and the plural marker are exclusive to these languages.

| Concept | Wayampi | Tekó | Zo'e | Borrowed from |
|---|---|---|---|---|
| Timbó liana | ɨmeku | beku | mekũ | Wayana |
| Plural marker | kũ | kom | kã | Cariban language |
| Pan | patu | patu | tapimã | From Portuguese through Wayana |
| Milk | tile | direr | tɨ | Creole |
| Mirror | warua | waruwa | poroesake | Língua Geral |
| Knife | marija | mᵇaridʒe | boke | Wayana |
| Salt | sautu | sautu | jukɨt | From English through Wayana |
| 3ʳᵈ pl. | kupa | kupa | – | Cariban language |
| Hen | masakala | masakala | ɲarĩ | Wayana |

the Kuikuro, according to whom they came from the North, passing via the Araguaia river through the Karajá territories, entering the Xingu basin via the Suyá-Missú river [166]. However, there is no archaeological sign of such an entrance of a TG group in the Upper Xingu.

In conclusion, a thorough history of the formation and development of TG languages, including the distinction between vertical, in-family, and out-family horizontal transmission, is yet to be written, reviewing everything that has been proposed so far. A unified interdisciplinary theory must give weight to data from linguistics, archaeology, ethnology, as well as genetics and the approach proposed in this article collaborates towards such an enterprise. We must also consider that the presence of material that is not vertically transmitted does not mean only that a tree will be distorted: it also means that even a "perfect" tree, one correctly capturing all relationships of descent, will mirror only a part of the history, especially if the spread of "horizontal" innovations was much faster than that of the "vertical" descent. A tree of lexical innovations is not a narrative of the history of the languages involved, but a means to tell one.

Although a critical review of the entire radiocarbon record associated with the TG dispersal is beyond the scope of this work, the quick assessment of the earliest regional dates summarized in this paper illustrates the difficulty of conciliating archaeological and linguistic data. In the future, strict criteria of chronometric hygiene should be applied to the published TG chronology to ascertain the reliability of each date [167]. For now, even if the long chronology proposed for some regions is discarded [70], the chronology available for the Paraná basin makes it difficult to argue for a recent arrival in the region. Numerous sites in southern Brazil and Argentina predate the second millennium [32], which is impossible to conciliate with an estimate of around 1750 years BP for the beginning of the TG dispersal to those regions.

In terms of phylogenetic studies based on linguistic data, besides incorporating expediencies from archaeology as priors, future work might investigate combining non-partial cognacy data with other features, such as partial cognacy sets, morphology, and phonology. For example, due to the strong composite character of TG lexicon, we decided not to use information on partial cognacy, despite its limited availability in [16]. Despite the source data carrying information on partial cognacy, we decided to employ exclusively simple cognates, also in consideration of how the substitution models available in Bayesian frameworks still demand nonstandard configurations to use them in an adequate way [81, 168].

Also deserving more consideration are TG practices that resulted in the conservancy of part of the language and its meanings observed in their material culture and environmental management. These are facts often historically, ethnographically, linguistically, and archaeologically recorded in different times and places by people with different expertise and objectives

who perceived various "empirical" and "theoretical" aspects of the TG peoples, as shown in [169]. Both ways lead to understanding the relations between the TG and other cultures, which included the appropriation and transformation of people, objects and language [170, 171], in processes characterized by "changes within continuities". The answer to these questions can be said to be the holy grail of TG historiography.

## Supporting information

**S1 File.**
(PDF)

## Acknowledgments

We thank the anonymous reviewers for their comments and suggestions that greatly helped us in improving our models, results, and manuscript. We thank Tatiana Merzhevich for helping in creating the figures in this paper.

## Author Contributions

**Data curation:** Fabrício Ferraz Gerardi, Carolina Coelho Aragon, Stanislav Reichert.

**Formal analysis:** Fabrício Ferraz Gerardi, Tiago Tresoldi.

**Supervision:** Fabrício Ferraz Gerardi, Tiago Tresoldi.

**Writing – original draft:** Fabrício Ferraz Gerardi, Tiago Tresoldi, Carolina Coelho Aragon, Stanislav Reichert, Jonas Gregorio de Souza, Francisco Silva Noelli.

**Writing – review & editing:** Fabrício Ferraz Gerardi, Tiago Tresoldi, Carolina Coelho Aragon, Stanislav Reichert, Jonas Gregorio de Souza, Francisco Silva Noelli.

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
