## [Decision Letter · Decision Letter 0]

19 Jan 2022

PONE-D-21-34420Lexical Phylogenetics of the Tupí-Guaraní Family: Language, Archaeology, and the Problem of ChronologyPLOS ONE

Dear Dr. Ferraz Gerardi,

Thank you for submitting your manuscript to PLOS ONE. After careful consideration, we feel that it has merit but does not fully meet PLOS ONE’s publication criteria as it currently stands. Therefore, we invite you to submit a revised version of the manuscript that addresses the points raised during the review process.

We look forward to receiving your revised manuscript.

Kind regards,

Søren Wichmann, PhD

Academic Editor

PLOS ONE

Journal Requirements:

2. We note that Figure(s) 1 and 3 in your submission contain [map/satellite] images which may be copyrighted. All PLOS content is published under the Creative Commons Attribution License (CC BY 4.0), which means that the manuscript, images, and Supporting Information files will be freely available online, and any third party is permitted to access, download, copy, distribute, and use these materials in any way, even commercially, with proper attribution. For these reasons, we cannot publish previously copyrighted maps or satellite images created using proprietary data, such as Google software (Google Maps, Street View, and Earth). For more information, see our copyright guidelines: http://journals.plos.org/plosone/s/licenses-and-copyright.

1. You may seek permission from the original copyright holder of Figure(s) 1 and 3 to publish the content specifically under the CC BY 4.0 license.  

3. We note that you have referenced (ie. Bewick et al. [5]) which has currently not yet been accepted for publication. Please remove this from your References and amend this to state in the body of your manuscript: (ie “Bewick et al. [Unpublished]”) as detailed online in our guide for authors

Reviewers' comments:

Reviewer's Responses to Questions

**Comments to the Author**

1. Is the manuscript technically sound, and do the data support the conclusions?

Reviewer #1: No

Reviewer #2: Yes

Reviewer #3: Partly

2. Has the statistical analysis been performed appropriately and rigorously? 

Reviewer #1: No

Reviewer #2: Yes

Reviewer #3: Yes

3. Have the authors made all data underlying the findings in their manuscript fully available?

Reviewer #1: Yes

Reviewer #2: Yes

Reviewer #3: Yes

4. Is the manuscript presented in an intelligible fashion and written in standard English?

Reviewer #1: No

Reviewer #2: Yes

Reviewer #3: Yes

5. Review Comments to the Author

Reviewer #1: In this paper, the authors apply Bayesian phylogenetic methods to the

Tupí-Guaraní language family in an attempt to illuminate the origin and

expansion of this family. The authors conduct a range of analyses (phylogenetic

network, phylogenetic dating, phylogeographic analysis) and conclude that the

origin of this family was around 2000 years ago.

The authors are to be commended for making all their data and analyses

available. This has not, unfortunately, been the case for previous work on

Tupí-Guaraní, and I admire the authors' good practice here.

In general, I like the paper, and think it has a lot of potential, but as it

currently stands it has some serious analytical flaws and the manuscript is

poorly structured. These shortcomings make it unclear what analyses were done

and why, and what the results actually mean. In particular I have some serious

concerns about the dating of the family. These flaws preclude its publication

in its current state and I therefore recommend *major revisions*.

My recommendations and suggestions for revision are listed below.

Major issues:

1. The phylogenetic analysis is problematic.

1a. how was the ascertainment correction implemented? The XML file in the

single partition analysis specifies that the first character in the alignment

is the ascertainment correction, and this character is all zero (correct).

However, there are 183 characters in the site alignment that are also empty.

This will severely affect the likelihood calculation and is likely to

over-inflate the tree height (this is why the Bouckaert et al. Indo-European

paper had to issue a correction https://doi.org/10.1126/science.1219669).

1b. Please check the calibrations. The paper lists 11 historical calibrations

and says that Tupinambá has a uniform calibration. The XML lists 4 calibrations

and specifies a normal distribution.

1c. There is no detail on what phylogeographic model was used, how it was

analysed, and what the results were (beyond a nice visualisation). More detail

please.

2. I am unconvinced by the dating of TG in these analysis.

First, there are very few historical calibrations and all of them are very

shallow calibrations on the order of a few hundred years. So, the young age of

TG here (compared to say, glottochronological estimates of ~3000 years) could

just be an artefact of the lack of timing information in the mid parts of the

tree.

Second, there is no attempt to assess the robustness of this timing to

different models. Root ages can vary massively between models.

Third, there is no discussion of the uncertainty in these estimates. They are

solely reported as "about". The median and 95% HPD interval (or standard

deviation or range) need to be reported. Looking at the log files, the root age

has a 95% HPD between 800 years and ~8000 years. This is quite a big range.

Given that the dating here is one of the major findings, I recommend that the

authors:

2a. add deeper calibrations if possible. The manuscript mentions a number of

potential calibrations later in the paper (e.g. L261-275 etc) which could,

hopefully, be used.

2b. investigate if the date estimates are robust across models. It is general

practice in phylogenetics to compare the fit of a range of candidate models

with a formal model comparison and discuss the variation in critical parameters

across these models. The authors should do this to fit with best practice (e.g.

covarion + strict clock, CTMC + strict, CTMC + relaxed). With the low timing

information in this analysis I suspect a strict clock might be a better fit to

the data and might, paradoxically, give a more robust estimate of the age. At

the very least, reporting the age estimates across a range of models will allow

the authors to be clear in what they're claiming. The paper by Kolipakam et al

(https://doi.org/10.1098/rsos.171504) might provide a good exemplar.

2c. If no other calibrations are available, then the authors could -- if they

wished to defend that date more -- conduct a formal test of whether their data

has sufficient temporal information to make inferences e.g. TempEst

(https://academic.oup.com/ve/article/2/1/vew007/1753488) or BETS

(https://doi.org/10.1093/molbev/msaa163)

3. It is unclear what analyses were done and *why*. The methods section lists

many different methods and talks about two or three ways of doing the same

thing, e.g.:

"These first models included both models generated procedurally, models

manually build with BEAUTi (a graphical user-interface for building models),

and models generated with beastling [84], also supported by simpler methods

such as UPGMA [85] and NJ [86].

... but the UPGMA and NJ results are not presented. Why use both beastling and

BEAUTi when they do essentially the same thing? the authors present one result

set -- but which one is it? the XML looks like a beastling file.

This is made more confusing by the authors discussing three(?) datasets: a

'main' dataset, a 'swadesh' dataset, and a 'rodrigues' dataset. Only the

results for 'main' are presented, which suggests that the other two datasets

were not used?

I recommend that the paper be streamlined to focus on a core set of analyses,

with a clear logic e.g. We wanted to examine the data for borowing and

conflicting signal, so we applied NeighborNet, we then wanted to infer the

relationships and timing so we...

4. The introduction does not work.

The ms. opens with the importance of linking archaeology and linguistics, but

doesn't do so (I would also argue that the biggest game changer in terms of

connecting linguistics to prehistory has been genetics and ancient DNA). The

paper then makes a throw-away statement about previous TG studies being

problematic, but doesn't really talk about why. The paper then moves onto

discuss the importance of TG, before coming back to archaeology.

I recommend deleting the first section completely and opening with the more

prominent section on why TG is important. The information about archaeology and

linguistics is important but fits more naturally later on with the introduction

of the archaeological findings.

5. Previous work needs more emphasis.

The introduction raises the issue about how problematic the previous

computational work on TG has been. This could do with more detail -- it is the

motivation behind this manuscript. More importantly TG is an interesting case

study where we have a few unrelated research groups working on the same family

coming up with different results.

I recommend the authors make more of this in the introduction to highlight the

differing studies. And then in the results discussion, the paper should compare

and contrast the authors results with the previous work. In short: why should I

prefer these results over the 3(?) or more previous studies?

Minor suggestions:

- Some of the figures have quite low quality. Please check the resolution.

- Figure 2 is not a distribution of TG archaeological sites. Please check

figure numbering.

- abstract: 'relevance' - relevance to what?

- footnote 2 should move into main text. It is important for the logic of the

argument in that section.

- The abbreviation NN for NeigborNet is not necessary.

- L95-97 is redundant, delete.

- L140-142 is cryptic. Explain.

- L185 data -> date

- L191, strictly speaking an MCC tree is not a "consensus" tree, it is a

summary tree.

- figure 4: 'reticular' is unusual.

- figure 4: it would help if the network was colored and labelled (groups I,

II, etc) to match the tree and help in the discussion section.

- footnote 5. Parts of this footnote explaining what a ẟ-score means should be

intext, however, the statement that quartets are the boxes on the NeighborNet

is incorrect.

- Appendix B: "ISO639P3code" is opaque to most people. The formal name is

ISO-639-3.

- Appendix C: "NeighborNet Metrics", the metrics in this table are not

technically linked to NeighborNets, so just call them Phylogenetic Metrics.

Also, please use the correct name ẟ-score rather than 'delta score'.

Reviewer #2: The manuscript “Lexical Phylogenetics of the Tupí-Guaraní Family: Language, Archaeology, and the Problem of Chronology” written by Gerardi et al. discusses much debated problem of Tupí-Guaraní expansion.

The authors correctly take in account current early radiocarbon dates related to ceramics associated to Tupí-Guaraní language groups, and they also make correct reference to some historical and ethnographic information of these groups. In general, the authors utilize quite comprehensive linguistic database of cognate data, that is, importantly, open for everyone. In their actual analysis they use Bayesian phylogenic methods in order to build Tupí-Guaraní expansion model with corresponding chronological estimations. Finally, the results are compared and contrasted to archaeological data that is summarized at the beginning of the study.

I am not specialist of lexical phylogenetics, but the supporting information, figures and tables help considerable the reading, even though, I must admit, the quality of Figures 1 and 2 are poor. Also specific uncertainties and the inclusion of special cases such as Omagua and Kokama are well explained - making the analysis convincing. The authors propose, in the text, a major split at ca. 950 BP dividing Tupí-Guaraní languages in two major groups, (1.) the southern group in southern Brazil, Paraguay, Bolivia and Argentina, and (2) all other Amazonian TG languages and Tupinambá divided into four clades (pages 9-10). Nevertheless, in the Abstract the split between Southern and Northern varieties is dated as 1100 BP.

In general, the authors announce “the importance of developing an interdisciplinary unified model incorporating data from both disciplines (archaeology and linguistics).” Undoubtedly this is a good objective even though not easy to achieve. First of all, one should remember that only rarely can a one-to-one relationship between the tangible and intangible evidence be demonstrated. Linguistically or historically recorded social, religious, political or linguistic changes do not immediately affect all material culture which can be detected archaeologically, or vice versa; a rapid change in material culture does not necessarily imply a simultaneous reorganization of social, religious, political or linguistic life (e.g. Braudel 1980:25–54, 64–82). In numerous cases, the lack of correlation between archaeological and intangible evidence has been documented (e.g. Pärssinen & Siiriäinen 1997; Marsh et al 2017). Also in Mexico, Michael E. Smith (1987) analyzed ethnohistorical and archaeological records of the Aztec expansion and concluded that the supposed artefactual markers of conquest spread to some provincial regions before the actual incorporation of these regions into the Aztec state. In a similar vein, Thomas Charlton (1981) has demonstrated that shifts from one period to another do not necessarily occur simultaneously in Mexican historical and archaeological sequences. Hence, before archaeologists and linguistics have agreed the exact meaning of concepts they are using, records of different disciplines should be analyzed separately to yield independent conclusions before correlation is attempted. As Smith (1987) proposed: “When the two records [of different disciplines] are compared, one should not confuse any resulting composite models with the independent primary data sets.” In other words, one should take in account the comparability of the data sets (so called Galton´s problem), and to be careful when using analogically models of other discipline. As Max Black (1962) once said: “Any would-be scientific use of an analogue model demands independent confirmation. Analogue models furnish plausible hypotheses, not proofs.”

After pointing these multi- and interdisciplinary problems, I must congratulate the authors of current manuscript to separate their linguistics results from early C14-dates obtained by archaeologists during their excavations. It is not surprising that finally the authors point out “the difficulties in reconciling archaeological and linguistic data.” In fact, even though the conservative behavior of historical Tupí-Guaraní in their material and economic cultures (p.12) can be accepted a posteriori, it does not mean that we may freely apply this statement analogically, a priori, to the situation around 2000 BP. Recently Pärssinen (2021) has published evidence from the Brazilian Acre, that in the Upper Purus corrugated, finger-nail and some polychrome decoration in grog tempered ceramics, often associated to Tupí-Guaraní styles, existed already 2000 calBP mixed with ceramics associated with so called Arawakan styles - all pointing to some kind of multicultural entity that built geometrically patterned ceremonial centers called “geoglyphs” in the region. Interestingly, also Pärssinen criticizes too keen association of material remains to linguistic groups when referring to pan-Amazonian traditions. Similarly with Anna Roosevelt and Denise Schaan he prefers to return to more neutral term such as Polychrome Horizon, and in fact, divides the Horizon into two phases, the Early (ca. 300 BC – 300/500 AD) and the Late Polychrome Horizon (ca. 900 – 1550 AD). Nevertheless, even though both Horizons were multicultural, he admits that that the first Horizon may partially be related to Arawakan expansion and the second one to Tupí-Guaraní expansion (see also, e.g. Almeida & Moraes 2016; Almeida & Neves 2014). Interestingly, this last period based on various C14 dating, seems to correlate quite well with the main results of the manuscript of Gerardi et al.

To conclude, I consider the manuscript of Gerardi et al. well written and its´ results to be an important contribution for multidisciplinary Tupí-Guaraní Studies. It also contains points of general interest, and in general, it satisfied all the publication criteria established by PLOS ONE. Hence, I recommend its´ publication with minor modifications.

References cited:

Almeida, F.O. & Moraes, C.P. 2016. A cerâmica polícroma do rio Madeira, in Cerâmicas arqueológicas da Amazônia: Rumo a uma nova síntese. Edited by C. Barreto, H. P. Lima, & C. Jaimes Betancourt, pp. 402-413. Belém: Iphan, Museu Paraense Emílio Goeldi.

Almeida, F.O. & Neves, E.G. 2014. The polychrome tradition at the Upper Madeira river, in Antes de Orellana. Actas del 3er Encuentro Internacional de Arqueología Amazónica. Edited by S. Rostain, v. 37, pp. 175-182. Quito: Instituto Francés de Estudios Andinos.

Black, M.1962. Models and Metaphors: Studies in Language and Philosophy. Ithaca, New York: Cornell University Press.

Braudel, F. 1980. On History. Translated by Sarah Matthews. London: Weidenfeld and Nicolson.

Charlton, T. H. 1981. Archaeology, Ethnohistory, and Ethnology: Interpretive Interfaces. Advances in Archaeological Method and Theory 4:129–176.

Marsh, Erik J., Ray Kidd, Dennis Ogburn & Victor Durán (2017). Dating the Expansion of the Inca Empire: Bayesian Models from Ecuador and Argentina. Radiocarbon 59:1, 117–140.

Pärssinen, M. 2021. Tequinho Geoglyph Site and Early Polychrome Horizon 300 BC - AD 300/500 in the Brazilian State of Acre. Amazônica: Revista de Antropologia, 13/2021(1), 177-220. DOI: http://dx.doi.org/10.18542/amazonica.v13i1.9095

Pärssinen, M. & A. Siiriäinen. 1997. Inka-style ceramics and their chronological relationship to the Inka expansion in the southern Lake Titicaca area (Bolivia). Latin American Antiquity, 8(3): 255–271.

Smith, M.E. 1987. The expansion of the Aztec empire: A case study in the correlation of diachronic archaeological and ethnohistorical data. American Antiquity 52(1): 37 – 54.

Reviewer #3: 1) I recommend the authors explicitly report the number of concepts (i.e., 183) used in the study they report, as it is far lower than the 447 they mention currently in text. I recommend they address whether they feel this dataset is of sufficient size to yield a sufficiently resolved tree (see Michael and Chousou-Polydouri 2019 for discussion); see also point (3).

2) I recommend the authors specifically state whether their dataset consists of root-meaning sets (see Chang et al. 2015 for this term) or cognate sets, that is, whether they have accounted for semantic shifts in determinations of cognacy. This is especially important for languages like Omagua and Kukama that have undergone relatively higher degrees of semantic shift, and more generally for the independence of the evolution of form and meaning.

3) I am concerned about the lack of explicit discussion of the widely differing posterior probabilities in Figure 5. Four of the 13 most specifically named clades have posterior probabilities of 0.67 or less (i.e., IIIb = 0.67, IVa = 0.60, IVb = 0.35, Ve = 0.58); and a fifth purported clade (Vd) is not monophyletic, a tacit claim the authors need to correct. Zooming out to the basic five named clades, III and IV are supported only with 0.65 and 0.66 posterior probabilities, respectively, and the separation of both of them from II is supported with a posterior probability of only 0.29! (Only I and V are supported with high posterior probabilities.) In fact, if one collapses all nodes with posterior probabilities lower than 0.80 -- the standard used by Michael et al. (2015), a valuable point of comparison based on 543 concepts instead of 183 -- most of the resolution/articulation in the tree in Figure 5 disappears. (As Michael and Chousou-Polydouri 2019 point out, even 0.80 is a rather permissive cutoff point; 0.90 or 0.95 would be more conservative.) On this view, while some important named clades remain (albeit with an internally more rake-like structure) -- such as I, IIIc, IVc, V -- most of the tree collapses into a massive rake, with the exception of some lower-level nodes such as that including Nheengatú, Omagua, and Kukama. (As an aside, it's striking that a node like IIIa, which Michael et al. recover with a posterior probability of 0.97, is here recovered only with a posterior probability of 0.38. I wonder if this is due to not properly identifying Omagua and Kukama cognates that have undergone semantic shift; see above.) Finally, note that the mentioned Amazonian-Southern split, which is rhetorically prominent in the authors' discussion, is recovered with a posterior probability of only 0.66. I recommend the authors provide an alternative, conservatively reported tree in which all nodes with posterior probabilities of less than 0.80 are collapsed, to facilitate comparison with extant literature (see next point).

4) The authors devote most of their discussion to engagement with Rodrigues and Cabral (2002) and related scholarship. As a computational phylogenetic study, it is rather surprising that theirs does not engage directly with the sole prior computational phylogenetic study of the family (Michael et al. 2015), systematically comparing the topology of their tree -- including differences in posterior probabilities -- with that of Michael and colleagues. It's all the more noteworthy since several of Rodrigues and Cabral's subgroups are defined by shared retentions (a point that Rodrigues himself makes in his 1984/1985 article), which are not valid criteria for subgrouping. The current study's generally low posterior probabilities, combined with the specious criteria for Rodrigues and Cabral's subgroups, results in two relatively weak points of comparison for the discussion. I recommend the authors reorient the discussion to engage with Michael et al. (2015).

5) Finally, the authors' chronological claims -- in particular that the entire TG family is a little over 1,000 years old -- are highly implausible. As early as Lathrap (1970), it was relatively clear that Tupian archaeological sites on the upper Amazon date to around 1100CE. This almost certainly corresponds to the expansion of pre-proto-Omagua-Kukama up the Amazon, a language that subsequently underwent massive restructuring (see Michael 2014) and was already differentiated into two languages by the 17th century. Yet in this study, proto-Omagua-Kukama is only about 250 years old! The authors open with a good overview of wide-ranging archaeological chronologies, and I recommend they improve the article by engaging with how its dates are significantly incommensurate not only with those archaeological chronologies but also textual documentation of some languages (e.g., Omagua) from the early 18th century (see Michael and O'Hagan 2016).

6. PLOS authors have the option to publish the peer review history of their article (what does this mean?). If published, this will include your full peer review and any attached files.

Reviewer #1: No

Reviewer #2: **Yes: **Martti H. Parssinen

Reviewer #3: No

---

## [Author Response · Author response to Decision Letter 0]

16 Jun 2022

Reviewer #1: In this paper, the authors apply Bayesian phylogenetic methods to the Tupí-Guaraní language family in an attempt to illuminate the origin and expansion of this family. The authors conduct a range of analyses (phylogenetic network, phylogenetic dating, phylogeographic analysis) and conclude that the origin of this family was around 2000 years ago.

The authors are to be commended for making all their data and analyses available. This has not, unfortunately, been the case for previous work on Tupí-Guaraní, and I admire the authors' good practice here.

In general, I like the paper, and think it has a lot of potential, but as it currently stands it has some serious analytical flaws and the manuscript is poorly structured. These shortcomings make it unclear what analyses were done and why, and what the results actually mean. In particular I have some serious concerns about the dating of the family. These flaws preclude its publication in its current state and I therefore recommend *major revisions*.

We thank the reviewer for seeing potential in the work and for commending the data availability, which we agree has unfortunately not been the case in the field (in a practice that, fortunately, seems to be rapidly changing within the context of open access and FAIR data). As for analytical flaws, we understand their position and believe the manuscript is much improved in this new submission, as we were able to account for the many valid suggestions and requests, along with trimming down the scope of the research being presented, which was one of the reasons for the admittedly lacking structure.

1a. how was the ascertainment correction implemented? The XML file in the single partition analysis specifies that the first character in the alignment is the ascertainment correction, and this character is all zero (correct). However, there are 183 characters in the site alignment that are also empty. This will severely affect the likelihood calculation and is likely to over-inflate the tree height (this is why the Bouckaert et al. Indo-European paper had to issue a correction https://doi.org/10.1126/science.1219669).

The reviewer was correct in pointing to the existence of empty characters in the alignment, which were due to a problem in our data preprocessing related to the order of filtering. This has been corrected following the practices described in the literature.

1b. Please check the calibrations. The paper lists 11 historical calibrations and says that Tupinambá has a uniform calibration. The XML lists 4 calibrations and specifies a normal distribution.

The calibrations listed in the manuscript referred to potential calibrations, most of which would not impact significantly the inferred tree as they were under a hundred years. We understand our wording was confusing and this has been corrected.

The issue is not relevant to the analysis anymore, as we decided to use a single date calibration (for the root, as described in the manuscript) and tip dates only for adjusting the length of the branches were it was applicable (given that we used the BD model).

1c. There is no detail on what phylogeographic model was used, how it was analysed, and what the results were (beyond a nice visualisation). More detail please.

We extended the description of the phylogeographic model in the manuscript, citing the relevant papers and the software used. The XML model was also included in the supplementary material.

2. I am unconvinced by the dating of TG in these analysis. First, there are very few historical calibrations and all of them are very shallow calibrations on the order of a few hundred years. So, the young age of TG here (compared to say, glottochronological estimates of ~3000 years) could just be an artefact of the lack of timing information in the mid parts of the tree.

We agree with the reviewer, and the new results, with a refined model, are closer to the glottochronological estimates. However, we understand this is an intrinsic problem with the phylogenetic dating of this family, considering there are no “fossil” data beyond Tupinambá and Old Guarani, and especially how we decided, by design, not to use archeological or genetic data a priori as potential calibrations.

Our design was to infer a good dated tree from the linguistic data (as events of lexical innovation) alone first, laying ground for future work with more linguistic material and, potentially, calibrations from linguistic evidence. The only arguably not exclusively linguistic calibration is the uniform distribution for the root age, which considers all the dates so far proposed in the different fields in a wide range of lower and upper boundaries, and the monophyletic constraints (especially for Mawé and Awetí), amply supported by the literature, which allow the algorithm to more easily “anchor” in a part of the timeline and to a very limited extent compensate the uncertainty given by the otherwise lack of calibrations save for the tip dates of the two “fossil” languages. We’d like to point that the new results considerably agree both with earlier glottochronological and with archeological hypotheses.

Second, there is no attempt to assess the robustness of this timing to different models. Root ages can vary massively between models.

We agreed with the reviewer, and had not performed such an assessment due to the confusion we could see by applying to the different models we were presenting. As we decided to reduce our research to a single study (dropping the “rodrigues” trees), we believe the manuscript is now clearer, and we assess the robustness comparing binary covarion on different datasets, both running with strict and relaxed log normal clocks. We believe this contributes more with the evaluation of the results, as common discussions with colleagues tend to center on the list of concepts to use.

Third, there is no discussion of the uncertainty in these estimates. They are solely reported as "about". The median and 95% HPD interval (or standard deviation or range) need to be reported. Looking at the log files, the root age has a 95% HPD between 800 years and ~8000 years. This is quite a big range.

We agree with the reviewer that the results were not properly discussed, and that the HPD interval was rather large. We are now reporting the 95% HPD of the most important splits, providing the summary trees in the appendix, which are much improved from the previous runs.

2a. add deeper calibrations if possible. The manuscript mentions a number of potential calibrations later in the paper (e.g. L261-275 etc) which could, hopefully, be used.

We thank the reviewer for the suggestion, but as discussed for this study we prefer to lay ground by making data and models openly available and not using non linguistic evidence in calibrations, also in order to bring forward the discussion with colleagues from archeology, anthropology, and genetics. This has been a good decision, in our view, as the results based exclusively on linguistic evidence (save for the generous prior for the uniform in the root) are largely agreeing with the dates suggested by archeology (which we intend to use in future studies, after considering the reception by the community).

2b. investigate if the date estimates are robust across models. It is general practice in phylogenetics to compare the fit of a range of candidate models with a formal model comparison and discuss the variation in critical parameters across these models. The authors should do this to fit with best practice (e.g. covarion + strict clock, CTMC + strict, CTMC + relaxed). With the low timing information in this analysis I suspect a strict clock might be a better fit to the data and might, paradoxically, give a more robust estimate of the age. At the very least, reporting the age estimates across a range of models will allow the authors to be clear in what they're claiming. The paper by Kolipakam et al (https://doi.org/10.1098/rsos.171504) might provide a good exemplar.

We ran covarion models with both a strict and a relaxed clock, as CTMC models were deemed too simple in their assumptions to model a family history as non-uniform as TG. We did, however, check the logmarginal likelihood of our different models with nested sampling, reporting the results in the supplementary material. It is necessary to note that we did non choose to report as our main result those with the best likelihood, for the reasons detailed in the manuscript when discussion those.

2c. If no other calibrations are available, then the authors could -- if they wished to defend that date more -- conduct a formal test of whether their data has sufficient temporal information to make inferences e.g. TempEst (https://academic.oup.com/ve/article/2/1/vew007/1753488) or BETS (https://doi.org/10.1093/molbev/msaa163)

We tested both datasets, with results reported at the individual concept and overall level, in terms of their tree-likeliness using TIGER scores. The test confirmed what is a consensus in the field (that TG languages don’t cluster neatly in lexical terms due to a high rate of non-vertical transmission), as reported in the manuscript and supported by the referenced literature.

This is made more confusing by the authors discussing three(?) datasets: a 'main' dataset, a 'swadesh' dataset, and a 'rodrigues' dataset. Only the results for 'main' are presented, which suggests that the other two datasets were not used?

We agree that our discussion and pipeline was confusing or, at least, not sufficiently explained. We decided to conduct a single study (equivalent to the “main” one of the initial submission), reporting an additional one (equivalent to “swadesh”), and leaving the other ones in plan for future work. We agree that this made the exposition of the results much clearer and easy to follow, and thank the reviewer for pointing it.

3. It is unclear what analyses were done and *why*. The methods section lists many different methods and talks about two or three ways of doing the same thing, e.g.:

We believe we have addressed all the issues pointed by this third item, as the models and analyses have changed. We also reviewed the text to make it clearer what was done and why done.

4. The introduction does not work. The ms. opens with the importance of linking archaeology and linguistics, but doesn't do so (I would also argue that the biggest game changer in terms of connecting linguistics to prehistory has been genetics and ancient DNA). The paper then makes a throw-away statement about previous TG studies being problematic, but doesn't really talk about why. The paper then moves onto discuss the importance of TG, before coming back to archaeology.

I recommend deleting the first section completely and opening with the more prominent section on why TG is important. The information about archaeology and linguistics is important but fits more naturally later on with the introduction of the archaeological findings.

As mentioned, this was stemming from the decision to organize data and reference for studying the family, also for other research groups, while only using linguistic evidence at first in light of long-standing disagreements between archeological schools and, especially, linguistic, anthropological, and archeological dating.

We do not think there is need to justify why TG is important beyond what is presented in the manuscript. The information on archaeology, summarizing the current knowledge, is intended to give an introduction to the dating issue, which has consequences for the results of linguistic analyses. Regarding DNA, the only study really relevant for our results is cited where appropriate (Silva et al. 2020). We agree with the reviewer that “ the biggest game changer in terms of connecting linguistics to prehistory has been genetics and ancient DNA”, but unfortunately there is not much available for TG populations.

5. Previous work needs more emphasis.

The introduction raises the issue about how problematic the previous computational work on TG has been. This could do with more detail -- it is the motivation behind this manuscript. More importantly TG is an interesting case study where we have a few unrelated research groups working on the same family coming up with different results.

I recommend the authors make more of this in the introduction to highlight the differing studies. And then in the results discussion, the paper should compare and contrast the authors results with the previous work. In short: why should I prefer these results over the 3(?) or more previous studies?

We expanded our consideration of previous studies, but stressed that it is difficult to compare with other quantitative analyses as the data and models of those studies are not open. We did compare the main different topological differences between our results and the previous ones (just as we compared with the main groups of linguistic and archeological studies) as much as necessary, exposing the reason for some of the disagreements. The discussion is supported by a large review on the literature on the expansion of TG, as evidenced by the numerous references on the matter.

Minor suggestions

We addressed most of the suggestions.

We thank the reviewer for the comments and suggestions that helped us in greatly improving our analyses.

Reviewer #2: The manuscript “Lexical Phylogenetics of the Tupí-Guaraní Family: Language, Archaeology, and the Problem of Chronology” written by Gerardi et al. discusses much debated problem of Tupí-Guaraní expansion.

The authors correctly take in account current early radiocarbon dates related to ceramics associated to Tupí-Guaraní language groups, and they also make correct reference to some historical and ethnographic information of these groups. In general, the authors utilize quite comprehensive linguistic database of cognate data, that is, importantly, open for everyone. In their actual analysis they use Bayesian phylogenic methods in order to build Tupí-Guaraní expansion model with corresponding chronological estimations. Finally, the results are compared and contrasted to archaeological data that is summarized at the beginning of the study.

I am not specialist of lexical phylogenetics, but the supporting information, figures and tables help considerable the reading, even though, I must admit, the quality of Figures 1 and 2 are poor. Also specific uncertainties and the inclusion of special cases such as Omagua and Kokama are well explained - making the analysis convincing. The authors propose, in the text, a major split at ca. 950 BP dividing Tupí-Guaraní languages in two major groups, (1.) the southern group in southern Brazil, Paraguay, Bolivia and Argentina, and (2) all other Amazonian TG languages and Tupinambá divided into four clades (pages 9-10). Nevertheless, in the Abstract the split between Southern and Northern varieties is dated as 1100 BP.

In general, the authors announce “the importance of developing an interdisciplinary unified model incorporating data from both disciplines (archaeology and linguistics).” Undoubtedly this is a good objective even though not easy to achieve. First of all, one should remember that only rarely can a one-to-one relationship between the tangible and intangible evidence be demonstrated. Linguistically or historically recorded social, religious, political or linguistic changes do not immediately affect all material culture which can be detected archaeologically, or vice versa; a rapid change in material culture does not necessarily imply a simultaneous reorganization of social, religious, political or linguistic life (e.g. Braudel 1980:25–54, 64–82). In numerous cases, the lack of correlation between archaeological and intangible evidence has been documented (e.g. Pärssinen & Siiriäinen 1997; Marsh et al 2017). Also in Mexico, Michael E. Smith (1987) analyzed ethnohistorical and archaeological records of the Aztec expansion and concluded that the supposed artefactual markers of conquest spread to some provincial regions before the actual incorporation of these regions into the Aztec state. In a similar vein, Thomas Charlton (1981) has demonstrated that shifts from one period to another do not necessarily occur simultaneously in Mexican historical and archaeological sequences. Hence, before archaeologists and linguistics have agreed the exact meaning of concepts they are using, records of different disciplines should be analyzed separately to yield independent conclusions before correlation is attempted. As Smith (1987) proposed: “When the two records [of different disciplines] are compared, one should not confuse any resulting composite models with the independent primary data sets.” In other words, one should take in account the comparability of the data sets (so called Galton´s problem), and to be careful when using analogically models of other discipline. As Max Black (1962) once said: “Any would-be scientific use of an analogue model demands independent confirmation. Analogue models furnish plausible hypotheses, not proofs.”

The idea of an interdisciplinary unified model is questioned that only rarely can a one-to-one relationship between the evidence tangible and intangible. We were not totally clear in our exposition, as we did not in any way propose what the reviewer understood of mechanically comparing the data, but to develop a unified interdisciplinary model with analytical capacity to use all available theories and information. We improved the clarification of such a model proposal. While the reviewer's afterthought is correct, it is necessary to remember that a linguistic family is in question, therefore, it is about the unification of information within the same set where there is a common cultural structure, produced from a reasonably well-known matrix. There is a corpus of data converging on several topics of interdisciplinary interest that need to be explained with perspectives quite different from the ingrained custom of producing research from a single science. We also decided to stress that this was the first time that such an approach was applied to TG, as it is part of a different way of building language trees which only recently has started to be applied to other language families as well (including much well studied ones as Indo-European and Sino-Tibetan).

After pointing these multi- and interdisciplinary problems, I must congratulate the authors of current manuscript to separate their linguistics results from early C14-dates obtained by archaeologists during their excavations. It is not surprising that finally the authors point out “the difficulties in reconciling archaeological and linguistic data.” In fact, even though the conservative behavior of historical Tupí-Guaraní in their material and economic cultures (p.12) can be accepted a posteriori, it does not mean that we may freely apply this statement analogically, a priori, to the situation around 2000 BP. Recently Pärssinen (2021) has published evidence from the Brazilian Acre, that in the Upper Purus corrugated, finger-nail and some polychrome decoration in grog tempered ceramics, often associated to Tupí-Guaraní styles, existed already 2000 calBP mixed with ceramics associated with so called Arawakan styles - all pointing to some kind of multicultural entity that built geometrically patterned ceremonial centers called “geoglyphs” in the region. Interestingly, also Pärssinen criticizes too keen association of material remains to linguistic groups when referring to pan-Amazonian traditions. Similarly with Anna Roosevelt and Denise Schaan he prefers to return to more neutral term such as Polychrome Horizon, and in fact, divides the Horizon into two phases, the Early (ca. 300 BC – 300/500 AD) and the Late Polychrome Horizon (ca. 900 – 1550 AD). Nevertheless, even though both Horizons were multicultural, he admits that that the first Horizon may partially be related to Arawakan expansion and the second one to Tupí-Guaraní expansion (see also, e.g. Almeida & Moraes 2016; Almeida & Neves 2014). Interestingly, this last period based on various C14 dating, seems to correlate quite well with the main results of the manuscript of Gerardi et al.

To conclude, I consider the manuscript of Gerardi et al. well written and its´ results to be an important contribution for multidisciplinary Tupí-Guaraní Studies. It also contains points of general interest, and in general, it satisfied all the publication criteria established by PLOS ONE. Hence, I recommend its´ publication with minor modifications.

We would like to thank the reviewer for praising our work and especially our proposal of an interdisciplinary approach bringing together linguistics and archeology in phylogenetic studies. The reviewer's remarks helped us improve the paper, and we received them as proof that the dialogue between the disciplines that we have been proposing is a profitable one.

#############################################################

Reviewer #3: 1) I recommend the authors explicitly report the number of concepts (i.e., 183) used in the study they report, as it is far lower than the 447 they mention currently in text. I recommend they address whether they feel this dataset is of sufficient size to yield a sufficiently resolved tree (see Michael and Chousou-Polydouri 2019 for discussion); see also point (3).

We made our methodology for selecting concepts clearer. With the expansion of TuLeD, now with 650 concepts for the next release (data was provided by the authors), we decided, after the suggestion of the reviewer, to include more concepts in the “full” analysis. We note, however, that there are many indications in the literature that the amount of concepts does not necessarily play such a significant role in the classification (see Kessler 2001, Brown et al. 2008, Gray et. al. 2009, Grollemund et al. 2015, Rama and Wichmann 2018, Kassian et al. 2021). The use of Michael and Chousou-Polydouri (2019), while consistent in its augmentation, is not very helpful in this regard since we cannot study or reproduce the analysis in Micheal et al. (2015).

The selection of concepts involved in the study was not arbitrary or subjective: we first took all the concepts in the lexical resource, dropped those without enough coverage among the languages (as described in the manuscript), and finally removed those that were considered problematic in terms of phylogenetic signal, and which the linguistic experts deemed of low data or evolutionary quality due to issues related to data collection (such as known problems in the elicitation) or due to their internal history in the family (such as foreign concepts known to have been internally borrowed or, at times, suspected of being borrowed from European languages via an intermediate, non-TG, native language). In the end, our criteria still selected over 400 concepts for the main analyses and 132 for the Swadesh analyses (out of 650), which are listed in the supplementary material.

Brown, C. H., Holman, E. W., Wichmann, S., & Velupillai, V. (2008). Automated classification of the world′ s languages: a description of the method and preliminary results. Language Typology and Universals, 61(4), 285-308.

Gray, R. D., Drummond, A. J., & Greenhill, S. J. (2009). Language phylogenies reveal expansion pulses and pauses in Pacific settlement. science, 323 (5913), 479-483.

Grollemund, R., Branford, S., Bostoen, K., Meade, A., Venditti, C., & Pagel, M. (2015). Bantu expansion shows that habitat alters the route and pace of human dispersals. Proceedings of the National Academy of Sciences, 112 (43), 13296-13301.

Kessler, B. (2001). The significance of word lists. Chicago University Press.

Kassian, A. S., Zhivlov, M., Starostin, G., Trofimov, A. A., Kocharov, P. A., Kuritsyna, A., & Saenko, M. N. (2021). Rapid radiation of the inner Indo-European languages: an advanced approach to Indo-European lexicostatistics. Linguistics, 59 (4), 949-979.

Taraka Rama and Søren Wichmann (2018) Towards identifying the optimal datasize for lexically-based Bayesian inference of linguistic phylogenies Proceedings of COLING 2018, Santa Fe.

2) I recommend the authors specifically state whether their dataset consists of root-meaning sets (see Chang et al. 2015 for this term) or cognate sets, that is, whether they have accounted for semantic shifts in determinations of cognacy. This is especially important for languages like Omagua and Kukama that have undergone relatively higher degrees of semantic shift, and more generally for the independence of the evolution of form and meaning.

We agreed with the suggestion and made sure to specify they are root-meaning sets, as the basic evolutionary character are lexical innovations.

3) I am concerned about the lack of explicit discussion of the widely differing posterior probabilities in Figure 5. Four of the 13 most specifically named clades have posterior probabilities of 0.67 or less (i.e., IIIb = 0.67, IVa = 0.60, IVb = 0.35, Ve = 0.58); and a fifth purported clade (Vd) is not monophyletic, a tacit claim the authors need to correct. Zooming out to the basic five named clades, III and IV are supported only with 0.65 and 0.66 posterior probabilities, respectively, and the separation of both of them from II is supported with a posterior probability of only 0.29! (Only I and V are supported with high posterior probabilities.) In fact, if one collapses all nodes with posterior probabilities lower than 0.80 -- the standard used by Michael et al. (2015), a valuable point of comparison based on 543 concepts instead of 183 -- most of the resolution/articulation in the tree in Figure 5 disappears. (As Michael and Chousou-Polydouri 2019 point out, even 0.80 is a rather permissive cutoff point; 0.90 or 0.95 would be more conservative.) On this view, while some important named clades remain (albeit with an internally more rake-like structure) -- such as I, IIIc, IVc, V -- most of the tree collapses into a massive rake, with the exception of some lower-level nodes such as that including Nheengatú, Omagua, and Kukama. (As an aside, it's striking that a node like IIIa, which Michael et al. recover with a posterior probability of 0.97, is here recovered only with a posterior probability of 0.38. I wonder if this is due to not properly identifying Omagua and Kukama cognates that have undergone semantic shift; see above.) Finally, note that the mentioned Amazonian-Southern split, which is rhetorically prominent in the authors' discussion, is recovered with a posterior probability of only 0.66. I recommend the authors provide an alternative, conservatively reported tree in which all nodes with posterior probabilities of less than 0.80 are collapsed, to facilitate comparison with extant literature (see next point).

The issues raised by the reviewer are to a large extent irrelevant now that the model and results have changed, including the posterior support in most cases. We also expanded our discussion on posterior values, but don’t believe they should be given that much preeminence in the interpretation. Some splits with full posterior support are obviously a consequence of monophyletic constraints specified directly or indirectly (in ours as in other studies), and low posteriors do not necessarily indicate worse results, even more so in the case of a language family which is known to carry conflicting phylogenetic signals due to borrowings, events of hybridization, a strong dialectal chain etc. A high posterior indicates that the mathematical optimization has converged in the search space, but not necessarily that it is “historically” correct – in an extreme case, if we set a monophyletic constraint that is wrong, the posterior will nonetheless be 1.0. We understand that a low posterior is not necessarily a bad result, as it might be just a numerical representation, an actual lower confidence, a non-tree-like evolutionary event that deserves further investigation or a data issue, perhaps with other methods. (see Kolipakam et al. 2018, Sagart et al. 2019).

Kolipakam, V., Jordan, F. M., Dunn, M., Greenhill, S. J., Bouckaert, R., Gray, R. D., & Verkerk, A. (2018). A Bayesian phylogenetic study of the Dravidian language family. Royal Society open science, 5(3), 171504.

Sagart, L., Jacques, G., Lai, Y., Ryder, R. J., Thouzeau, V., Greenhill, S. J., & List, J. M. (2019). Dated language phylogenies shed light on the ancestry of Sino-Tibetan. Proceedings of the National Academy of Sciences, 116(21), 10317-10322.

4) The authors devote most of their discussion to engagement with Rodrigues and Cabral (2002) and related scholarship. As a computational phylogenetic study, it is rather surprising that theirs does not engage directly with the sole prior computational phylogenetic study of the family (Michael et al. 2015), systematically comparing the topology of their tree -- including differences in posterior probabilities -- with that of Michael and colleagues. It's all the more noteworthy since several of Rodrigues and Cabral's subgroups are defined by shared retentions (a point that Rodrigues himself makes in his 1984/1985 article), which are not valid criteria for subgrouping. The current study's generally low posterior probabilities, combined with the specious criteria for Rodrigues and Cabral's subgroups, results in two relatively weak points of comparison for the discussion. I recommend the authors reorient the discussion to engage with Michael et al. (2015).

We chose not to engage with Michael et al. (2015) in terms of model and data, as (with the risk of being repetitive) these are not public. However, we now consider the topological differences between our results and theirs, highlighting the main disagreements and discussing them from both a phylogenetic and general linguistic perspective. This has improved our work, as the engagement with this previous study was indeed important.

We reconstructed the engagement with Rodrigues and Cabral (2002), also considering how their groupings are to large extent based on phonology and shared retentions (as correctly pointed how by the reviewer) and how experts tend to point a difference between lexical and phonological clades in TG. We decided to expand the engagement with Michael et al. (2015) and especially with archeological proposals.

5) Finally, the authors' chronological claims -- in particular that the entire TG family is a little over 1,000 years old -- are highly implausible. As early as Lathrap (1970), it was relatively clear that Tupian archaeological sites on the upper Amazon date to around 1100CE. This almost certainly corresponds to the expansion of pre-proto-Omagua-Kukama up the Amazon, a language that subsequently underwent massive restructuring (see Michael 2014) and was already differentiated into two languages by the 17th century. Yet in this study, proto-Omagua-Kukama is only about 250 years old! The authors open with a good overview of wide-ranging archaeological chronologies, and I recommend they improve the article by engaging with how its dates are significantly incommensurate not only with those archaeological chronologies but also textual documentation of some languages (e.g., Omagua) from the early 18th century (see Michael and O'Hagan 2016).

The updated analyses, considering what one of the reviewers suggested, improved the results, which now show that the TG family is 2500-1500 years old. We do agree with you (the reviewer) that an early date in the upper Amazon could correspond to the migration of the proto-Omagua-Kokama. The NN confirms the proximity of Omagua-Kokama to Nheengatu, despite a high amount of exclusive features in the first, which could be reason for the split date of the whole clade where they are placed in the MCC being potentially older than expected (when considering the archeological estimates for the arrival of the Tupinambá on the coast, even though we must consider, in phylogenetic terms, that the split involves the originate [i.e., Proto-Tupinambá] and not the branch tip [i.e., the Tupinambá recorded by Europeans in the 1500s]). Since the idea of the paper is to present the issue with dates (archaeology and linguistics) based on an interdisciplinary point of view,, we do intend in a future study to consider the Omagua-Kokama origin based, among other things, on Michael (2014) and Michael and O'Hagan (2016). This was not done in the present study, because we decided to employ such a kind of information, along with other constraints and calibrations (potentially coming from archeology), only in future studies using more complex phylogenetic methods. This approach would surely consider, for the case of Omagua and Kokama, the fact that they were already differentiated in the eighteenth century, both for the sake of a better date involving their split and the for the overall contribution this information (as many others we plan to use) would give to the tree inference. In the results we are presenting, due to the model we employed (also in face of the monophyletic constraints), this information would likely impact exclusively the split date of Omagua and Kokama, without any significant change to tree otherwise.

Thank you for your suggestions! I hope we have replied to all of them appropriately.

---

## [Decision Letter · Decision Letter 1]

15 Jul 2022

Lexical Phylogenetics of the Tupí-Guaraní Family: Language, Archaeology, and the Problem of Chronology

PONE-D-21-34420R1

Dear Dr. Ferraz Gerardi,

We’re pleased to inform you that your manuscript has been judged scientifically suitable for publication and will be formally accepted for publication once it meets all outstanding technical requirements.

Kind regards,

Søren Wichmann, PhD

Academic Editor

PLOS ONE

Additional Editor Comments (optional):

Reviewers' comments:

Reviewer's Responses to Questions

**Comments to the Author**

1. If the authors have adequately addressed your comments raised in a previous round of review and you feel that this manuscript is now acceptable for publication, you may indicate that here to bypass the “Comments to the Author” section, enter your conflict of interest statement in the “Confidential to Editor” section, and submit your "Accept" recommendation.

Reviewer #1: All comments have been addressed

Reviewer #3: All comments have been addressed

2. Is the manuscript technically sound, and do the data support the conclusions?

Reviewer #1: Yes

Reviewer #3: (No Response)

3. Has the statistical analysis been performed appropriately and rigorously? 

Reviewer #1: Yes

Reviewer #3: (No Response)

4. Have the authors made all data underlying the findings in their manuscript fully available?

Reviewer #1: Yes

Reviewer #3: (No Response)

5. Is the manuscript presented in an intelligible fashion and written in standard English?

Reviewer #1: Yes

Reviewer #3: (No Response)

6. Review Comments to the Author

Reviewer #1: I'm happy that the authors have addressed all my concerns.

........................................

Reviewer #3: (No Response)

7. PLOS authors have the option to publish the peer review history of their article (what does this mean?). If published, this will include your full peer review and any attached files.

Reviewer #1: No

Reviewer #3: No

---

## [Editor Report · Acceptance letter]

13 Apr 2023

PONE-D-21-34420R1 

Lexical Phylogenetics of the Tupí-Guaraní Family: Language, Archaeology, and the Problem of Chronology 

Dear Dr. Ferraz Gerardi:

I'm pleased to inform you that your manuscript has been deemed suitable for publication in PLOS ONE. Congratulations! Your manuscript is now with our production department. 

Kind regards, 

on behalf of

Dr. Søren Wichmann 

Academic Editor

PLOS ONE